# Temporal organization of stride-to-stride variations contradicts predictive models for sensorimotor control of footfalls during walking

Madhur Mangalam[1]*, Damian G. Kelty-Stephen[2], Joel H. Sommerfeld[1], Nick Stergiou[1,3], Aaron D. Likens[1]

**1** Division of Biomechanics and Research Development, Department of Biomechanics, and Center for Research in Human Movement Variability, University of Nebraska at Omaha, Omaha, NE, United States of America, **2** Department of Psychology, State University of New York at New Paltz, New Paltz, NY, United States of America, **3** Department of Department of Physical Education, & Sport Science, Aristotle University, Thessaloniki, Greece

☯ These authors contributed equally to this work.
* mmangalam@unomaha.edu

**Data Availability Statement:** All relevant data are within the manuscript and its Supporting information files.

## Abstract

Walking exhibits stride-to-stride variations. Given ongoing perturbations, these variations critically support continuous adaptations between the goal-directed organism and its sur-roundings. Here, we report that stride-to-stride variations during self-paced overground walking show cascade-like intermittency—stride intervals become uneven because stride intervals of different sizes interact and do not simply balance each other. Moreover, even when synchronizing footfalls with visual cues with variable timing of presentation, asyn-chrony in the timings of the cue and footfall shows cascade-like intermittency. This evidence conflicts with theories about the sensorimotor control of walking, according to which internal predictive models correct asynchrony in the timings of the cue and footfall from one stride to the next on crossing thresholds leading to the risk of falling. Hence, models of the sensori-motor control of walking must account for stride-to-stride variations beyond the constraints of threshold-dependent predictive internal models.

## Introduction

Human movement performance requires a subtle mixture of stability and adaptability. As the best musicians and athletes know, each performance depends on building solid foundations through repeated practice and an ongoing openness to new and often unforeseen perturba-tions. Even the most rigorous, repetitive training must allow the body to navigate unplanned perturbations and then to fold the inevitable deviations into a continuing trajectory toward task completion. Perfect replication is almost impossible, even with a lifetime of practice. More importantly, given the uncertainty of context, perfect replication might even be unwelcome, as it might hamper expert performance. Instead, expert performance might often exploit variabil-ity to continually tune movements and fit the situation. In this sense, variability is not an

**Funding:** This work was supported by the Center for Research in Human Movement Variability at the University of Nebraska at Omaha, funded by the NIH award P20GM109090. The funders had no role in study design, data collection and analysis, decision to publish, or preparation of the manuscript.

**Competing interests:** The authors have declared that no competing interests exist.

obstacle to stability but instead complementary support ensuring an ongoing adaptive fit between the organism and the environment [1–5].

Walking is a flagship example of how stability and adaptability complement each other. At first glance, walking appears strictly periodic, with an inverse pendulum of the upper body wielding two pendular legs below it, in antiphase. Indeed, periodic models of walking can explain a significant proportion of the stride-to-stride variations characteristic of walking [6–8]. These models decompose gait into a linear combination of independent oscillators whose contributions add together with minimal interactive interference. Of course, we expect all good models to have residuals because variability may exceed the modeled stability. However, standard practice assumes the "ergodicity" of the proposed components and their combined action. Consequently, these pendular dynamics among separable parts amount to an expectation that residuals will be uncorrelated, allowing the average trajectory to be representative across time and individuals.

Ergodicity pertains to the degree of representativeness in our summary descriptions of the behaviors we are interested in, across various sampling methods. A system is ergodic when the average of one individual exemplar of the system across time resembles the average for a sample of exemplars (Fig 1, ***top left***). Ergodicity is not plain stationarity, here meant to imply a constant mean ($M$), and standard deviation ($SD$) over time. While both ergodicity and stationarity imply the stability of $M$ and $SD$, unlike stationarity, ergodicity is directly concerned with the representative relationship between the individual case and the larger sample. Hence, a system can be stationary—that is, its $M$ and $SD$ remain stable over time—yet break ergodicity (Fig 1, ***top right***). Conversely, a non-stationary system will necessarily break ergodicity (Fig 1, ***bottom***). In gait, ergodicity amounts to the resemblance between individual strides and a time average of strides. Even stationary processes can fail to be ergodic [9–13]. Notably, while stationarity presumes the $M$ and $SD$ of stride intervals remain the same across many strides, ergodicity presumes any single stride is comparable to any other stride or predictable from any average.

## Ergodicity as a foundation for predictive models of gait control

Ergodicity and its failure become essential for predictive-model-based explanations of gait because the representativeness of a gait cycle or stride lies at the heart of all predictive models for sensorimotor control. One or more variants of these models are often referred to as internal models, forward models, internal forward models, or corollary inverse models [14–20]. Models, in general, formalize a process in bare forms that capture the fundamental, most reliable essence of the unfolding dynamics (e.g., the "template"), while the details of the movement can vary based on the walking environment (i.e., "anchor" [21]). Besides all quibbling over any internal agent or observer for whom a model "re-presents" information, modern explanations of gait rely on the premise of a representative, basic template for the control of gait. Whether written in terms of dynamics, kinematics, or task [22–24], the representativeness of this model is the foundation for any model supporting predictive control. The so-called noisiness of neurophysiology and its capacity for clogging the movement systems with delays in information transfer has been relentlessly cited as justifications for predictive models for sensorimotor control [14–20]. To be clear, we do not wish to say that the ergodicity of the gait dynamics and predictive control of gait are equivalent. We wish to assert that ergodicity is essentially a requirement of predictive control: prediction requires representativity, and representativity is not available if the more profound constraint of ergodicity fails to hold.

Control by predictive models respects a similar principle; let the pendular dynamics of the legs unfold undisturbed until deviation requires correction. The similarity of Newtonian

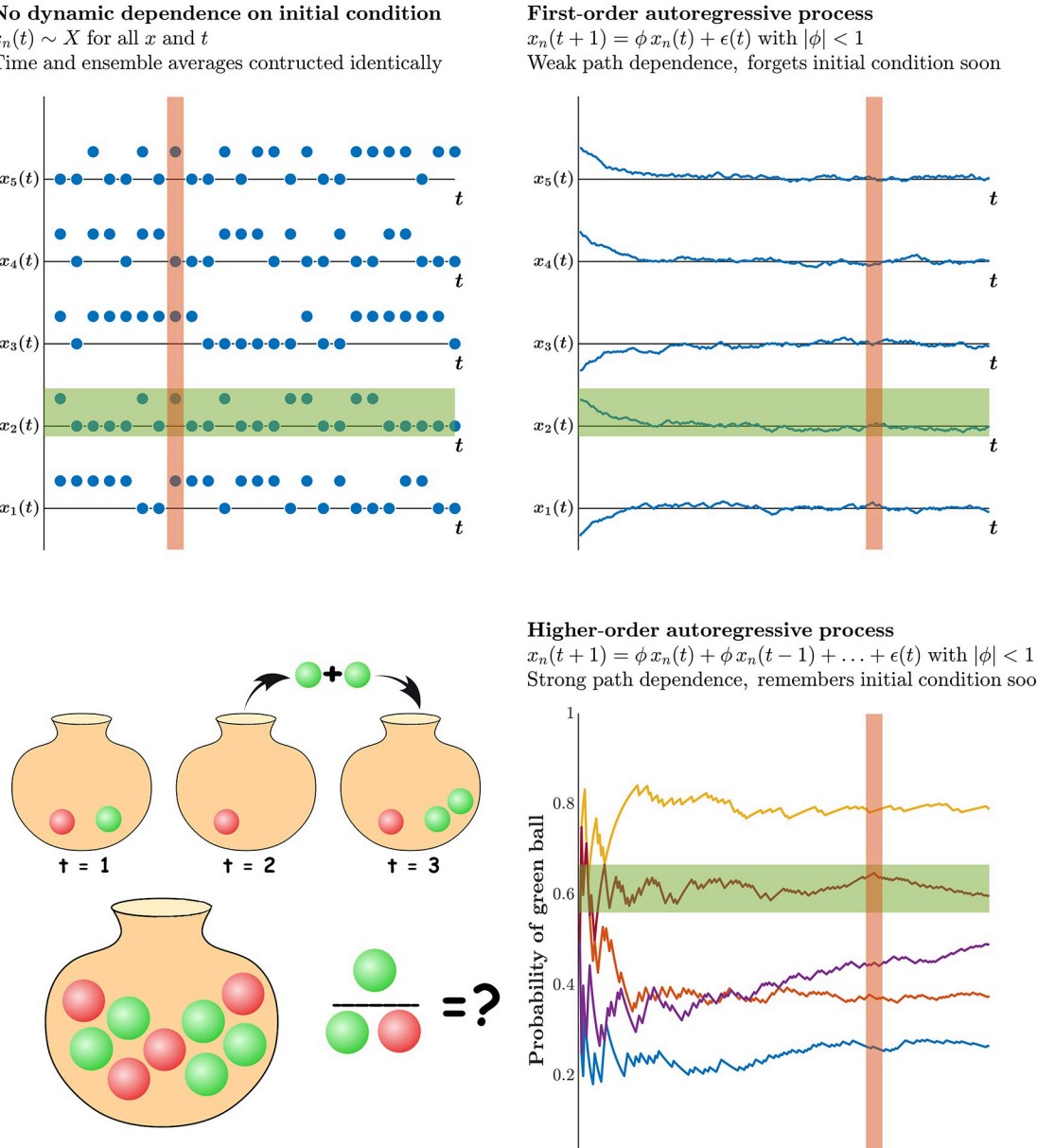

**Fig 1. Illustration of ergodic and nonergodic processes.** Tossing a fair coin multiple times in a sequence is an ergodic process (***top left***). In this case, the ensemble average across multiple coin-toss trajectories (red vertical patch) is equivalent to the time average for each and every single trajectory of coin toss (green horizontal patch). A first-order autoregressive process is also ergodic after a few initial points in the time series ***top right***. In this case as well, the ensemble average across multiple trajectories (red vertical patch) is equivalent to the time average for each and every single trajectory (green horizontal patch). In contrast, higher-order autogressive processes break ergodicity (***bottom***). In this example, a marble is taken out from a pot containing red and green marbles and an additional marble of the same color is added. Consequently, the probability of drawing a green ball diverges based on the initial few moves. In this case, the ensemble average across multiple trajectories (red vertical patch) differs from the time average for each and every single trajectory (green horizontal patch).

physics across all terrains allows all templates to assume similar rules for the harmonic oscillators enlisted for locomotion. Deviations require correction, but only those sufficient in magnitude to exceed some predetermined threshold. Predictive models contain certain limits, known as thresholds, which signify the maximum safe variation from the predicted outcome.

If the movement or change in a certain parameter exceeds these thresholds, then the risk of falling significantly increases. Essentially, these thresholds act as a safety measure, preventing any drastic or potentially dangerous deviations from the expected results. Predictive models are expressions of probabilities, and the thresholds are implicit statements about the likelihood of a fall. Below thresholds, movements are free to vary and thought to reflect the sum of many independent random sources of variance. Above the thresholds, movements are corrective interventions meant to reverse excessive deviation. Hence, the typical predictive model reflects a constrained form of variation. More technically, this theorizing often invokes control by a predictive model that blends expectations of uncorrelated additive white Gaussian noise (awGn) combined with short-lag, negatively autocorrelated corrections [14, 20].

The precise blending of uncorrelated awGn with negative short-lag autocorrelation is fundamental to information-processing views of movement coordination in general [25, 26]. Predictive modeling has now become an essential component of several theories related to movement coordination. This is because sensory input and corrections are processed through the neural structure that is characterized by noisy, lagging transmissions. In other words, predictive modeling allows modeling movement coordination despite these limitations in neural processing [27, 28]. Furthermore, confidence in the representativeness of the gait cycle goes hand in hand with the overall confidence that neuronal activity might be equally ergodic, equally representative from one action potential to another, and so delivering the same laggy, noisy contribution in all cases. Certainly, the evidence from nervous systems at rest is that neuronal dynamics are ergodic [29]. Fundamentally, the ergodicity required of the predictive model at the longer timescale of the task, is equally required at shorter timescales of single neuronal spikes.

Challenges to explanation by predictive models accrue from multiple sources. For instance, when locomotion begins, the neural events associated with footfalls are neither independent nor predictable [29], implying broken ergodicity [30–36]. Moreover, stride-to-stride variations are also not awGn but are more consistent with fractional Gaussian noise (fGn; awGn is a special case of fGn which lacks any temporal correlations) [37–39]. Of course, fGn can result from models that incorporate inertial tendencies into independent random sources of variance [40–44], and that could make predictive modeling more feasible (although see [45, 46]). However, despite attempts to generate a predictive model that would produce fGn in stride-to-stride variations (e.g., [6]), these attempts ignore two critical points. First, ergodicity breaks progressively as temporal correlations increase [9–11, 13]. This effect would make any average estimate (e.g., of prior performance over short or long time scales) progressively more unstable. This destabilization of average estimates would render such models less capable of projecting reliable predictions of fall risk, thus loosening their predictive grip on the actual behavior. Second, analysis of movement variations in numerous tasks suggests that evidence of fGn-like monofractality is only the tip of an iceberg, with multifractality—time varying fractal properties—playing a much larger role than previously thought [47, 48].

In order to deploy predictive models for gait, one must consider the various forms of unpredictable deviation from the mean. For instance, the awGn model involves deviations that converge on a mean, requiring minimal vigilance under the risk thresholds. On the other hand, the fGn model reflects a standard deviation that can grow at a rate faster than predicted by the central limit theorem. The term "monofractal" is used to describe a statistical scenario where a given time series can follow only one power-law exponent $H$ for the growth of standard deviation, and where the exponent $H$ may have a fractional value. However, analyses reveal that gait is actually multifractal, exhibiting multiple power-law exponents for the growth of standard deviation across various timescales and deviation sizes [49–57]. Consequently, predictive models that assume a monofractal scaling estimate of strides in gait (e.g., [6]) tend to systematically underestimate the actual multifractality of stride-to-stride variations.

Worse yet for predictive models, multifractal fluctuations suggest that a predictive model can not simply focus on one step at a time. Multifractal fluctuations arise from an interdependence across scales [47, 58, 59], emphasizing the risk of stride-to-stride variations breaking ergodicity [60–62]. The seemingly pendular regularity of gait could make this point feel strange. However, gait might break ergodicity but only weakly and not so strongly as to upset upright posture. Indeed, ergodicity breaking may be less of a dichotomy than previously imagined and offer more of a continuum. Biological functions are widely known to exhibit weak ergodicity breaking [63]. And we might see this weak ergodicity breaking through the lens of comparably weakened evidence of multifractality. Gait comprises strides whose series unfold across time with a statistical profile that nearly matches the conditions of fGn, for which estimatable power-law exponents are fewer, suggesting a less multifractal structure. The relatively narrower range of fractal exponents relating standard deviation to timescale in the gait stride time series supports an expectation that stride-to-stride variations produce weak ergodicity. Ergodicity breaking may be weak enough to invite predictive models, but it may be persistent enough to frustrate us with the ultimate unpredictability of individual strides.

As we have said, prediction needs ergodicity: predicting the next stride requires an uncorrelated, ergodic structure among strides. The empirical evidence—fractal and multifractal alike —of nonergodicity in stride-to-stride variations poses severe challenges for predictive models. First, monofractal or multifractal strains of nonergodicity will thwart a predictive model expecting uncorrelated deviations and entailing a negative response to superthreshold deviations. The premise for internal predictive models proposed to control gait is that deviations are uncorrelated in time and require only a corrective response to superthreshold deviations. This premise fails to hold in the empirical record of the stride-to-stride variations. Second, fractality in the stride-to-stride variations suggest that the deviations follow from long-past events in the fractal case. Likewise, multifractality in the stride-to-stride variations suggest that the deviations follow the interactions of events over multiple scales of the past in the multifractal case. Hence, we might enforce control as a negative short-lag autocorrelation to correct recent events above a risk threshold in predictive models. But this control will not address deviations from a longer-range fractal or multifractal process. Negative short-lag autocorrelations inherent to predictive models expecting uncorrelated strides provides an inadequate solution to deviations that unfold across scales.

We acknowledge the anticipatory aspect of movement coordination but doubt the necessity or feasibility of internal predictive models. We might have found explanations of prospective, anticipatory control of gait—and movement more generally—on a different foundation. The plain and simple fact is that our body shows a remarkable capacity to look forward in steering locomotion and adjusting to upcoming perturbations and threats to stability [64–70]. Some of the earliest investigations of behavioral synchrony with isochronous auditory cues found the mean synchronization error—asynchrony—was negative, indicating that participants do not simply react to the cue but tap before the cue onset [71]. Negative mean asynchrony has only continued to provide critical evidence for the predictive modeling: it can seem like we predict because we can respond before the cue—in gait [72–82] but also in movement more generally [83–85]. According to many of these accounts, maintaining synchrony requires prediction because of differences in sensory delays between receiving sensory feedback from tapping or stepping and sensory stimulation by the cue [72, 73, 86]. However, is anticipation necessarily from predictive modeling? The potential fragility of ergodicity at the neural, motoric, and task performance levels puts unforgiving constraints on this possibility. Denying anticipatory movement coordination seems as counter-productive as proposing internal predictive models could somehow do the mathematically impossible and predict an ergodic process in the raw measurements.

## Ergodicity as a foundation for linearization-based models of gait control, with or without prediction

Then again, the science of gait control is already well versed in the challenges of prediction and has wisely responded to these challenges with advanced modeling of gait control that depends less on the maybe-not-representable history and more on each current gait cycle [87, 88]. It might seem at first glance that our concern for ergodicity leaves these modeling strategies free from critique so long as they relax the requirement of prediction. However, we restate: that ergodicity is not the whole of prediction but is instead a broader concern that would be required to support prediction. Ergodicity extends beyond prediction, though, and is more generally the support for all models of gait that operate by linearization of the measured process. These relatively prediction-agnostic modeling strategies lean sooner on the parameter dynamics and have arrived at evolutionary optimal settings that distinguish stable from unstable manifolds to avoid the representativity of state dynamics [88]. Moreover, rather than leaving control mechanisms contingent on crossing state-dependent thresholds, the control mechanisms can be contingent on the contours of these distinct manifolds. This position is strategic because it attempts to invest its confidence not in the ergodicity of short-term behavior but in the ergodicity of a longer-term evolutionary path.

Nevertheless, we suspect that the ergodic requirements of this mode of control which is manifold-driven and is agnostic to prediction are no less firm, and we can identify at least three points where ergodicity breaking would pose challenges for even such prediction-lean modeling. First, the background commitment to the proposed evolutionary optimum runs aground recent theorizing that suggests the evolutionary record exhibits sufficient discontinuity, divergence, and emergence to thwart ergodic characterization (e.g., [89, 90]). Modern appeals to ergodicity in neurobiology within the organism lifespan report an expectation only of "local" ergodicity (e.g., [91])—an expectation that does not meet with extensive empirical support across the developmental trajectory [92]. On evolutionary time, the appeal to physical constraints of biological tissue and genetic coding can make an ergodic time scale available to empirical and theoretical work provided the domain is finite and discretely countable (e.g., genetic exploration involving a discrete alphabet of nucleotides and amino acids) [93]. When we consider the influence of ecological changes in the coevolutionary process, the present manifestations of ergodicity within stationary states along the evolutionary trajectory seem to obscure a dynamic internal structure of fluctuations. This concealed complexity is primed to unleash the next major advancement or catastrophe, intertwined with a cascade of interdependent factors (e.g., [94]). As a result, contemporary theories of evolutionary optimization strive to acknowledge the necessity of multiple potential wells with intricately patterned boundaries. In this scenario, achieving ergodicity necessitates embracing the existence of multiple possibilities, thereby negating any straightforward notion of ergodicity in its core essence. The evolutionary perspective thus can manage ergodic description at the cost of requiring Ptolemaic epicycles to describe nonergodic Keplerian truths [90]. At each point, at each time scale, there is a perpetual deferring of the ergodicity to a narrower view, for example, so that we may have ergodicity to suit at least the modeling frame under consideration but not ergodicity at all scales. Ultimately, the expectation of ergodicity in biological evolution appears sooner to be a convenience assumed without thorough empirical confirmation and without much empirical grounds for presuming it beyond the empirical record [95].

Indeed, for present purposes, the physical constraints of gait are explicit for modern humans and current ecological constraints. So the preceding caution about ergodicity breaking thwarting evolutionary optimality could be moot. Indeed, evolutionary optima are not fixed points but expressions of adaptivity, and we understand that the manifold-themed modeling

makes explicit room for the contextual variations that complicate the preceding issue of evolutionary optima. However, our second and third subsequent points remain, hinging on the fact that, even without explicit predictive mechanisms, the prediction-agnostic modeling strategies appealing to stable vs. unstable manifolds still operate upon the empirical gait dynamics. Our second point that parsing a system's phase space trajectory into stable vs. unstable manifolds rests on the ergodicity of the underlying measure [96]. Furthermore, this ergodic measure must be smooth everywhere for stable- vs. unstable-manifold modeling. The multifractal fluctuations regularly found in strides or asynchronies with metronome onsets [47, 49–57, 97] would not be sufficiently differentiable to qualify as smooth [98].

Our third and last point is that if the current stable- vs. unstable-manifold modeling is indistinguishable from uncontrolled manifold modeling (UCM; [99]), then it is concerning for ergodicity considerations that UCM is a linearization using the Jacobian matrix. Indeed, the Jacobian matrix makes the exact requirement of smoothness and bidirectional differentiability, as highlighted in our second point. However, the explicit recognition of equivalence among linearization strategies reaches a profound point about ergodicity. Linearization depends on a decomposition into separate constituent components whose independence is the reason they may be summed up to produce their model predictions [48, 92, 100]. Some independent parameters might be nonlinear, for example, harmonic, polynomial, or exponential, but general linear modeling refers to that broad class of strategies in which stable parameters sum together to produce the outcome [101]. Ergodicity is required to ensure that a sum will remain the same across space or time; the invariance of the summation is what allows us to generalize from one ensemble of linear models to a single linear process of the same form. It is ergodicity that allows us to generalize from an ensemble of trajectories along the uncontrolled manifold and guarantee successful task completion for an individual trajectory following the average of that ensemble. No matter whether gait control predicts, linearization is a neat compromise that presumes to bargain with the fundamental nonergodicity of perceiving acting systems. By assuming ergodicity where we cannot, this bargain surrenders any capacity we might have—as scientists—to predict over any but the shortest time scales. The sun is setting on the general linear model of nonergodic processes, and the alternatives warrant cogent consideration.

In summary, whether they involve explicit prediction, models that encode behavioral processes using linearization depend on ergodicity. We have only called specific attention to predictive models because they appeal to cognitive-psychological processes of anticipation. Anticipation is a power typically not attributed to physical systems—or at least not to the Newtonian mechanics attributed to gait. The physical constraints implicit in presumed evolutionary optima [88, 93] may soften the requirement for any psychological powers of anticipation. Nevertheless, we are attempting to open the discourse for introducing a chaotic sort of physics that does support anticipatory synchronization without a biological-evolutionary premise [102]. These more chaotic physical constraints afford higher-dimensional elaborations whose nonlinearity may thwart linearization and suggest a cascade-like mechanism affording nonlinear interactions across time scales that would break ergodicity and support anticipation. Thus, we might embody gait that both breaks ergodicity and has a route to anticipation. The ergodicity breaking would only reduce the applicability of gait control resting on linearization.

## Beyond ergodicity: Cascade-dynamical routes forward to anticipatory gait control

Fortunately, alternative proposals exist that embrace the so-called noisiness of neurophysiology and its incident delays in information transfer [103]. Exclusive foundation on predictive

models would make anticipation "weak" in the sense of being prone to limitations of a predictive model. Models of so-called "strong" anticipation have eschewed predictive models in favor of foundations in long-range, multi-scaled coupling amongst random fluctuations that could support more adaptive behavior [104–107]. Perceiving-acting organisms may use nonergodic support instead of predictive models. We see great promise in recent attempts to establish control systems on stochastic foundations [108–110]. Not all stochastic foundations will naturally do: awGn will not generate adaptive corrections to superthreshold deviations. But fractal and multifractal deviations may carry within themselves a solution to the problem of superthreshold deviations. Their long-range correlations are not simply a statistical novelty: they reflect a movement system rich in potentially cascade-like nonlinear interactions across timescales. Cross-scale interactions may support model-free anticipation more robust to ergodicity breaking [111]. In particular, those cascade-like processes generate multifractal nonlinearities missing in their corresponding surrogates [112, 113]. Surrogates are synthetic time series with an identical linear temporal structure to the original, but they are missing any of the nonlinearities present in the original time series. Importantly, such a multifractal nonlinearity supports dexterous movement coordination across time, even under unpredictable circumstances [1, 106, 114–117].

When we remove or diminish the capacity to predict subsequent strides, perceiving-acting organisms appear to respond to nonergodic and unpredictable environmental cues with nonergodic variations in their behavior [118–120]. They even tune the multifractality of movement variations to the multifractality of unpredictable contextual circumstances in other motor contexts [105, 106, 115]. It could be that the negative mean asynchrony we, as scientists, identify as anticipation in the locomoting perceiving-acting organism rests on a longer-term substrate of gait fluctuations that carry fractal and multifractal profiles. The anticipatory behavior we see and sometimes interpret as prediction may not be predictable and ergodic. However, it is possible that the interactions across timescales implicated in cascade dynamics could shepherd the short-term dynamics of each step through the longer-term undulations of variation as participant attention and engagement waxes and wanes across the temporal extent of the task. However, negative mean asynchrony might depend on the long-range—specifically multifractal—structure of nonergodic stride-to-stride variations.

Walking exhibits stride-to-stride variations. Given ongoing perturbations, these variations critically support continuous adaptations between the goal-directed organism and its surroundings. Here, we report that stride-to-stride variations during self-paced overground walking show cascade-like intermittency—stride intervals become uneven because stride intervals of different sizes interact across time and do not simply cancel each other out. Moreover, even when synchronizing footfalls with visual cues with variable presentation timing, we report that asynchronies between visual cues and footfalls show cascade-like intermittency. This evidence conflicts directly with theories about the sensorimotor control of walking, according to which internal predictive models correct asynchrony between cue and footfall only from one stride to the next on crossing thresholds leading to the risk of falling. Hence, models of the sensorimotor control of walking must account for stride-to-stride variations beyond the constraints of threshold-dependent predictive internal models.

We tested the above ideas in an experimental study of cued walking using visual cues with various temporal structures in healthy adults. Our experimental design allowed us to directly examine whether the temporal organization of stride-to-stride variations during overground walking contradicts the predictive models for sensorimotor control. We put forward the following hypotheses. First, we predicted that stride-to-stride variations would be nonergodic, especially in the unperturbed case of self-pacing; that is, individual stride intervals will not resemble the average of stride intervals over the long run (Hypothesis 1a). As participants

coordinated with cues bearing more of the nonergodic temporal structure (e.g., pink noise), we expected ergodicity breaking in stride interval time series (Hypothesis 1b) and in the time series of asynchronies between visual cues and footfalls (Hypothesis 1c). We also expected the extent of ergodicity breaking in both time series to depend on the extent of ergodicity breaking in the visual cues (Hypothesis 1d). Second, we predicted that the time series of stride intervals and asynchronies between visual cues and footfalls would be fractal (Hypothesis 2a) and multi-fractal (Hypothesis 2b) rather than identically distributed, randomized or independently sequenced noise with the same values and probability distribution. Third, we predicted that the multifractal nonlinearity in asynchronies between visual cues and footfalls would correlate with the negative mean asynchrony comparing footfalls to cues (Hypothesis 3).

## Results

This study involved ten participants completing five walking trials on a 200-meter indoor track (Fig 2). The first trial was self-paced walking (SPW). The next four required synchronizing footfalls with visual cues displayed on a small video screen mounted on eyeglass frames as an oscillating horizontal bar. Footswitch sensors detected heel-strike events. Bar-oscillation timing in the four trials was manipulated using different pacing signals, including pink noise pacing signal (PPS), shuffled pink noise pacing signal (SPPS), Gaussian distributed random pacing signal (GRPS), and uniformly distributed random pacing signal (URPS; Fig 3, **top**). In all four paced-walking conditions, the average and standard deviation of the timing of the

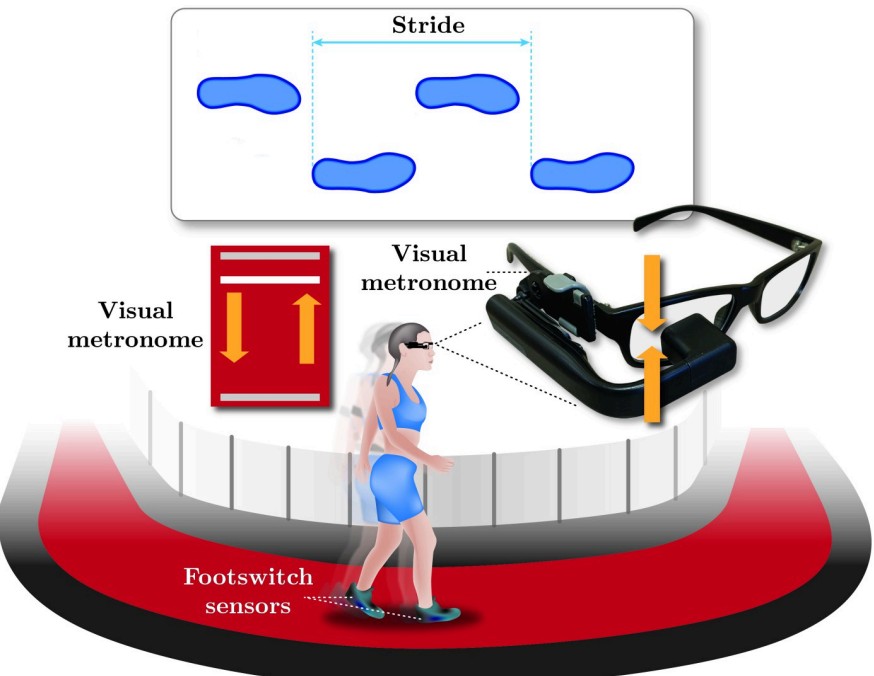

**Fig 2. Experimental setup.** Participants completed five overground walking trials on a 200 m indoor track. The first trial was self-paced walking (SPW). The next four trials required participants to time their right heel strikes with oscillations of a horizontal bar presented on a mini HDMI video screen mounted on eyeglass frames. Footswitch sensors placed under both heels identified heel-strike events. The timing of the oscillations in these four trials varied according to four signals: pink noise pacing signal (PPS), shuffled pink noise pacing signal (SPPS), Gaussian distributed random pacing signal (GRPS), and uniformly distributed random pacing signal (URPS). The order of the four paced-walking trials was randomized for each participant.

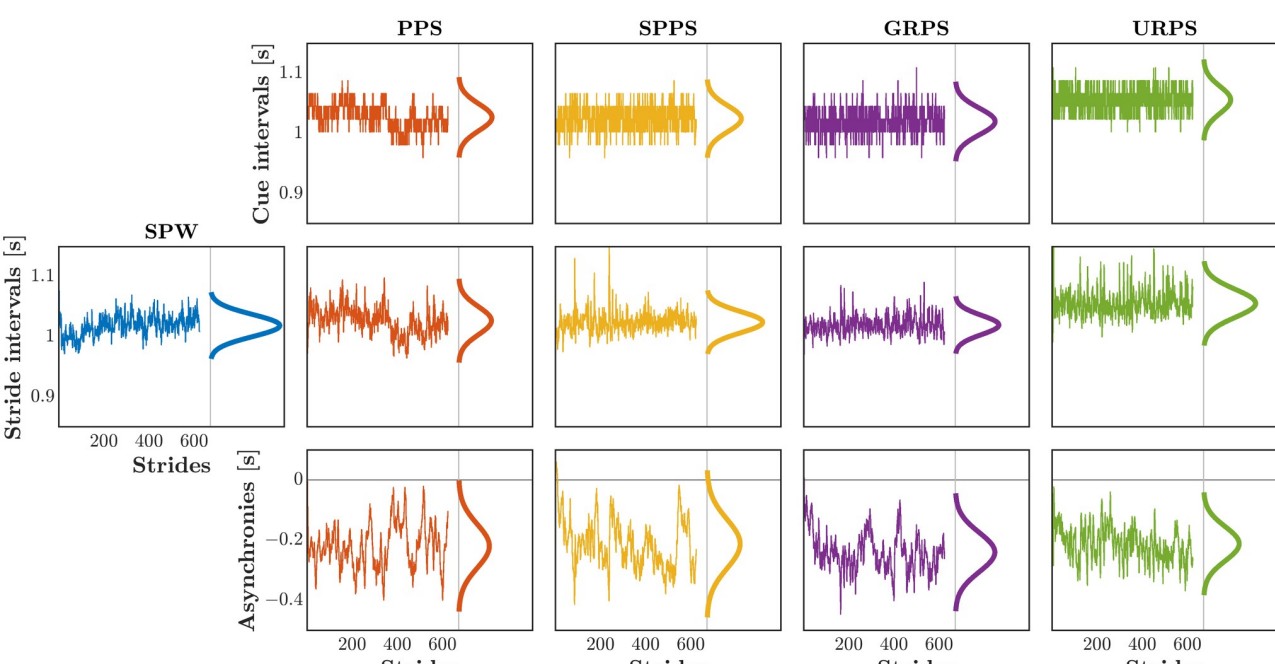

*Paced walking yielded negative mean asynchrony between visual cues and footfalls*

**Fig 3. Paced walking yielded negative mean asynchrony between visual cues and footfalls irrespective of the temporal structure of cue intervals.**
Time series and probability distributions of cue intervals (***top***), stride intervals (***middle***), and asynchrony in the timings of visual cues and footfalls (***bottom***) for a representative participant. Despite the deterministic temporal structure of the pink noise pacing signal (PPS) and nondeterministic or random structure of the shuffled pink noise pacing signal (SPPS), Gaussian distributed random pacing signal (GRPS), and uniformly distributed random pacing signal (URPS), mean asynchrony in the timings of visual cues and footfalls was always negative.

visual cues were adjusted to match each participant's average and standard deviation of stride-to-stride intervals recorded during SPW. Each trial resulted in three-time series of approximately 600 strides—namely the time series of (i) cues, (ii) footfalls, and (iii) asynchronies between visual cues and footfalls.

## Paced walking yielded negative mean asynchrony between visual cues and footfalls

Despite the homogeneity of variance in the pacing signals (URPS; Fig 3, ***top***), paced walking resulted in greater heterogeneity in stride-to-stride variations, as illustrated by stride intervals (URPS; Fig 3, ***middle***) and asynchronies between visual cues and footfalls (URPS; Fig 3, ***bottom***). Despite the stride-to-stride heterogeneity, participants successfully synchronized their footfalls with the visual cues. Across all pacing signals, participants timed their footfalls in anticipation of visual cues instead of the actual cueing events. Notably, we observed negative mean asynchrony, irrespective of the pink noise pacing signal's deterministic yet unpredictable temporal structure. Mean asynchrony did not differ across the four paced-walking conditions ($B \pm SE = 0.0078 \pm 0.0054$, $t_{38} = 1.4576$, $P = 0.15318$, 95%CI = [−0.0030, 0.0187]), showing no significant differences in negative value ($Ms \pm SDs = −150 \pm 50$ ms, $−143 \pm 49$ ms, $−141 \pm 60$ ms, and $−200 \pm 59$ ms for PPS, SPPS, GRPS, and URPS, respectively). Hence, negative mean asynchrony is not simply typical of coordination with isochronous cues [71–73, 76–79, 82] but may extend to coordination with temporally unpredictable cues.

Asynchronies between visual cues and footfalls broke ergodicity We quantified the ergodicity of each time series of cue intervals, stride intervals, and asynchronies between visual cues and footfalls using a dimensionless statistic of ergodicity breaking $E_B$, known as the Thirumalai-Mountain metric [121, 122]. $E_B$ is an inverse metric of ergodicity, reflecting how the long-range persistence failure of variance to converge, in effect, undermines (or "breaks") the representativity of a sample, thereby undermining ergodicity. This $E_B$ statistic subtracts the squared total-sample variance, $\langle \overline{\delta^2(x(t))} \rangle^2$, from the average squared subsample variance, $\langle [\overline{\delta^2(x(t))}]^2 \rangle$, and divides the resultant by the squared total-sample variance, $\langle \overline{\delta^2(x(t))} \rangle^2$:

$EB(x(t)) = \frac{\langle [\overline{\delta^2(x(t))}]^2 \rangle - \langle \overline{\delta^2(x(t))} \rangle^2}{\langle \overline{\delta^2(x(t))} \rangle^2}$; where $\delta^2(x(t))$ is sample-to-sample variance and this relationship is effectively the variance of the sample variance divided by the squared total-sample variance. As noted above, $E_B$ is inversely related to ergodicity. Ergodicity is evident as rapid decay of $E_B$ to 0 for progressively larger samples, that is, $E_B \to 0$ as $t \to \infty$. For instance, for Brownian motion $E_B(x(t))) = \frac{4}{3}\frac{\Delta}{t}$, where $\Delta$ is the time lag [123, 124] (also see [125]). Slower decay indicates less ergodic systems in which trajectories are less reproducible. No decay or convergence to a finite asymptotic value indicates strong ergodicity breaking [9]. Thus, $E_B$-vs.-$t$ curves allow testing whether a given time series fulfills or breaks ergodic assumptions and how strongly it breaks ergodicity.

Given the finite-size constraint on our samples, we do not expect convergence to zero but focus instead on the rate of decay in $E_B$ between original time series and shuffled counterparts as evidence of ergodicity breaking. Shuffled versions are apt for comparison because ergodicity hinges on whether an individual sequence exemplifies an average of sequences. Shuffling breaks sequence, producing additive white Gaussian noise (awGn) distributing independently around a mean. By the design of the study, the original pink noise pacing signal showed shallower decay in $E_B$ with $t$ compared to its shuffled counterpart $((E_B(x(t))) = 0.74\frac{\Delta}{t}$ and $1.09\frac{\Delta}{t}$, where $\Delta = 4$; Fig 4, *top*), suggestive of ergodicity breaking due to the long-range temporal structure [10, 11, 13] In contrast, for all other cue signals, $E_B$ was more suggestive of ergodicity, with original cue interval series showing rapid decay in $E_B$ with $t$ comparable to shuffled counterparts for SPPS $(E_B(x(t))) = 0.97\frac{\Delta}{t}$ and $1.19\frac{\Delta}{t}$), GRPS $(E_B(x(t))) = 1.00\frac{\Delta}{t}$ and $0.99\frac{\Delta}{t}$), and URPS $(E_B(x(t))) = 1.36\frac{\Delta}{t}$ and $1.11\frac{\Delta}{t}$).

Overall, stride intervals only showed ergodicity breaking in response to nonergodic visual cues (Fig 4, *middle*). There was a rapid decay in $E_B$ with $t$ comparable to its shuffled counterpart suggestive of ergodicity in the original stride interval time series during self-paced walking $(E_B(x(t))) = 0.95\frac{\Delta}{t}$ and $0.98\frac{\Delta}{t}$) and while synchronizing footfalls with both the Gaussian distributed random pacing signal $(E_B(x(t))) = 1.01\frac{\Delta}{t}$ and $1.01\frac{\Delta}{t}$) and uniformly distributed random pacing signal $(E_B(x(t))) = 0.92\frac{\Delta}{t}$ and $1.00\frac{\Delta}{t}$). However, synchronizing footfalls with the nonergodic pink noise and shuffled pink noise pacing signals broke ergodicity, resulting in a shallower decay of EB with t compared to its shuffled counterpart for both PPS $(E_B(x(t))) = 0.57\frac{\Delta}{t}$ and $1.00\frac{\Delta}{t}$) and SPPS $(E_B(x(t))) = 0.62\frac{\Delta}{t}$ and $1.18\frac{\Delta}{t}$).

Finally, irrespective of stride-interval ergodicity, asynchronies between visual cues and footfalls broke ergodicity across all pacing conditions (Fig 4, *bottom*), with the original time series showing a much shallower decay in $E_B$ with $t$ compared to their shuffled counterpart: pink noise pacing signal $(E_B(x(t))) = 38\frac{\Delta}{t}$ and $1.21\frac{\Delta}{t}$), shuffled pink noise pacing signal $(E_B(x(t))) = 0.07\frac{\Delta}{t}$ and $1.04\frac{\Delta}{t}$), Gaussian distributed random pacing signal $(E_B(x(t))) = 0.57\frac{\Delta}{t}$ and $1.01\frac{\Delta}{t}$), and uniformly distributed random pacing signal $(E_B(x(t))) = 0.27\frac{\Delta}{t}$ and $1.10\frac{\Delta}{t}$). That is, the average of all stride intervals was not representative of a typical stride and hence, is not amenable to predictive modeling and, likewise,

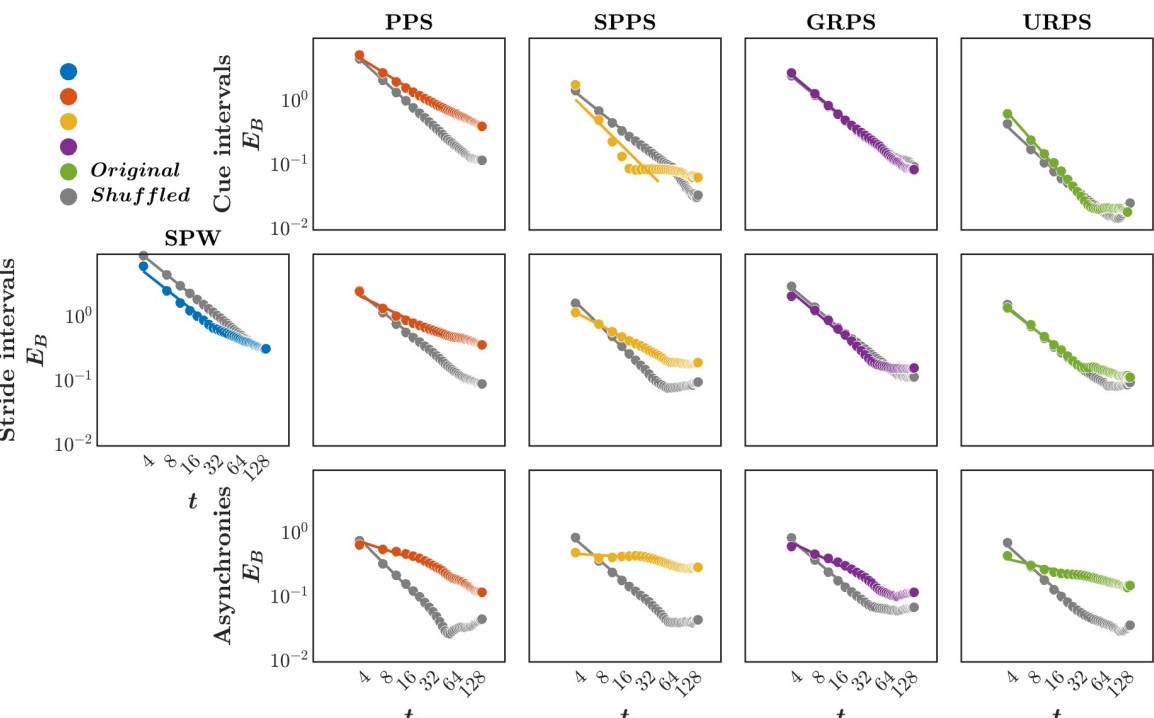

**Fig 4. Asynchronies between visual cues and footfalls broke ergodicity irrespective of ergodic properties of the cue intervals and stride intervals.** Log-log plots of ergodicity breaking factor $E_B$ vs. time $t$ for the time series of cue intervals (***top***), stride intervals (***middle***), and asynchronies between visual cues and footfalls (***bottom***) averaged across all participants. The stride intervals for self-paced walking (SPW) appeared ergodic, showing only a slightly shallower decline in $E_B$ vs. $t$ for the original time series as opposed to its shuffled version at medium to longer timescales. Cue intervals, stride intervals, and asynchronies between visual cues and footfalls broke ergodicity more clearly for the pink noise pacing signal (PPS) and shuffled pink-noise pacing signal (SPPS) with the shallow and steep decline of $E_B$-vs.-$t$ curves for the original and shuffled time series, respectively. In contrast, despite cue and stride intervals exhibiting ergodicity for the Gaussian distributed random pacing signal (GRPS) and uniformly distributed random pacing signal (URPS), asynchronies between visual cues and footfalls in both conditions broke ergodicity, although to a lesser extent for GRPS.

asynchrony between visual cue and footfall from one stride to the next is also not amenable to predictive modeling.

**Asynchronies between visual cues and footfalls showed persistence in variation.**
Despite its ergodicity, stride-to-stride variations during self-paced walking exhibited specific temporal structure. There were specifically correlations between present values of stride intervals with past stride intervals at several lags. The Hurst exponent, $H_{fGn}$, describes the gradual decay of correlations between stride intervals across longer separations in time (Fig 5). These temporal correlations resemble fractional Gaussian noise (fGn) whose power-law decay of autoregressive coefficient $\rho$ with lag $k$ as $\rho_k = \frac{1}{2}(|k+1|^{2H} - 2|k|^{2H} + |k-1|^{2H})$, and following $H_{fGn}$, the moments of the autocorrelation diverge for $0.5 < H_{fGn} \leq 1$. $H_{fGn}$ encodes the degree of long-range persistence ($0.5 < H_{fGn} < 1.0$; large values are followed by large values and vice versa) or long-range antipersistence ($0 < H_{fGn} < 0.5$; large values are followed by small values and vice versa). As stated above, time series with $H_{fGn} = 0.5$ or 1 can be considered as having the Fourier power spectrum consistent with "white noise" or "pink noise," respectively. Stride intervals in healthy adults exhibit long-range temporal correlations that fall into the pink noise classification [37, 38, 54], and aging and neuropathy lead to a loss of this pink noise structure

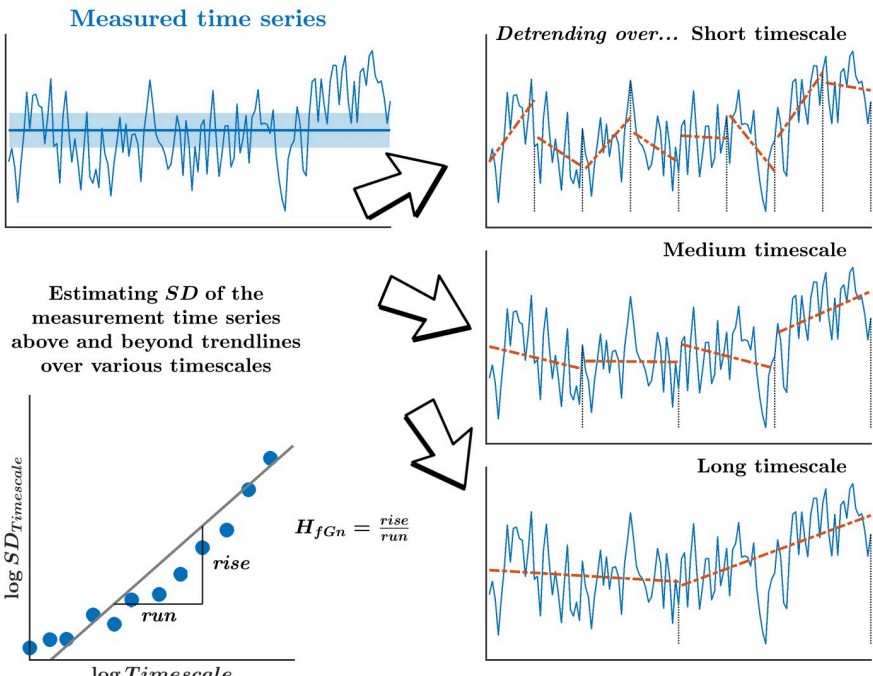

**Fig 5. Schematic portrayal of the Hurst exponent $H_{fGn}$.** The fractal-scaling exponent $H_{fGn}$ provides the first entree to describe cascade dynamics underlying our data. Specifically, $H_{fGn}$ relates how $SD$-like fluctuations grow across many timescales, encoding how the correlation among sequential measurements might decay slowly across longer separations in time. Detrending fluctuations over progressively longer timescales removes mean drift across each of these timescales.

[39, 126–128]. We computed $H_{fGn}$ using detrended fluctuation analysis [129, 130] to investigate whether stride interval time series and time series of asynchronies between visual cues and footfalls show long-range antipersistence or persistence. Both possibilities pose a challenge to predictive modeling accounts. Antipersistence ($0 < H_{fGn} < 0.5$) will imply a similar negative autocorrelation by which predictive control corrects superthreshold asynchrony, and that might seem less contradictory than the accumulation of asynchrony in the persistent case. However, the challenge is that fGn-type correlations are long-range, whereas predictive modeling's negative autocorrelation is strictly short-lag, limiting the predictive focus to just-previous footfalls.

By the design of the study, the pink noise pacing signal was highly persistent ($M \pm SD$ $H_{fGn}$ = 0.99 ± 0.08; $t_{9,2}$ = 15.3838, $P$ = 9.0569 × 10$^{-8}$, 95%CI = [0.4065, 0.5467]; Fig 6, **top**), whereas each of the other pacing signals was not persistent (SPPS: $H_{fGn}$ = 0.53 ± 0.02, $t_{9,2}$ = 2.2927, $P$ = 0.0476, 95%CI = [0.0005, 0.0772]; GRPS: $H_{fGn}$ = 0.53 ± 0.02, $t_{9,2}$ = −0.6806, $P$ = 0.5131, 95%CI = [−0.0372, 0.0200]; and URPS: $H_{fGn}$ = 0.51 ± 0.01, $t_{9,2}$ = −0.1704, $P$ = 0.8685, 95%CI = [−0.0313, 0.0269]). Self-paced walking showed persistence in stride intervals ($H_{fGn}$ = 0.86 ± 0.14; $t_{9,2}$ = 6.6569, $P$ = 9.2999 × 10$^{-5}$, 95%CI = [0.2331, 0.4731]), confirming past research [38, 131, 132]. Synchronizing footfalls with the PPS accentuated the persistence in stride intervals ($H_{fGn}$ = 0.97 ± 0.09, $B \pm SE$ = 0.1080 ± 0.0398, $t_{45}$ = 2.7165, $P$ = 0.0093, 95%CI = [−0.0279, 0.1881]; Fig 6, **middle**), but all other pacing signals prompted persistence to a lesser degree (SPPS: $H_{fGn}$ = 0.64 ± 0.084, $B \pm SE$ = −0.2216 ± 0.0398, $t_{45}$ = −5.5748, $P$ = 1.3295 × 10$^{-6}$, 95%CI = [−0.3017, −0.1416]; GRPS: $H_{fGn}$ = 0.64 ± 0.06, $B \pm SE$ = −0.2194 ± 0.0398, $t_{45}$ = −5.5189, $P$ = 1.6072 × 10$^{-6}$, 95%CI = [−0.2995, −0.1393]; and URPS: $H_{fGn}$ = 0.58 ± 0.07, $B \pm$

*Asynchronies between visual cues and footfalls showed long-range persistence*

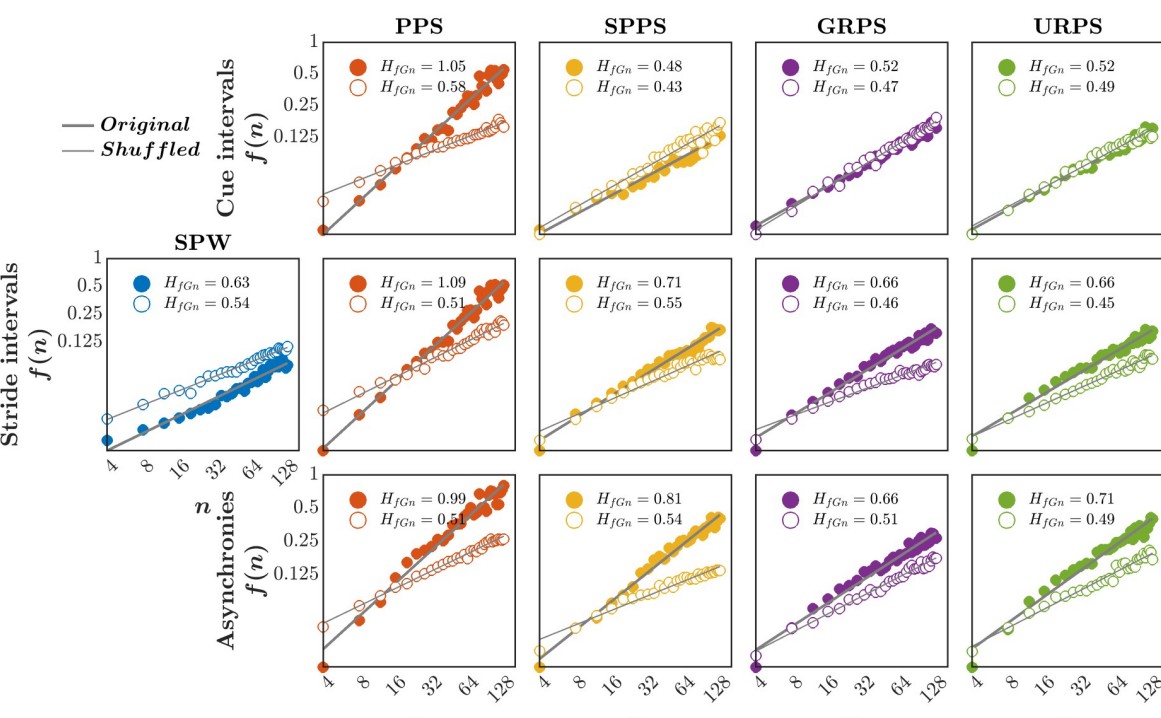

**Fig 6. Asynchronies between visual cues and footfalls showed persistence irrespective of persistence in cue or stride intervals.** The Hurst exponent, $H_{fGn}$, encoding the persistence in measured fluctuations is the slope of log-log plots of fluctuation function, $f(n)$, as a function of Bin size, $n$, in cue intervals (**top**), stride intervals (**middle**), and asynchronies between cues and footfalls (**bottom**) for a representative participant. The stride intervals for self-paced walking (SPW) showed strong persistence—$H_{fGn}$ close to 1 (thick vs. thin grey lines), indicating that large fluctuations are followed by large fluctuations and vice versa. Synchronizing footfalls with the pink noise pacing signal (PPS) accentuated the persistence in stride intervals while walking in synchrony with the shuffled pink noise pacing signal (SPPS), Gaussian distributed random pacing signal (GRPS), and uniformly distributed random pacing signal (URPS) abated the persistence in stride interval—$H_{fGn}$ smaller than 1 but still greater than 0.5, despite all these latter cue signals being completely uncorrelated across time, i.e., $H_{fGn}$ close to 0.5. Contrary to the prediction of strong short-lag antipersistence following the predictive modeling accounts of stride-to-stride control of stepping), asynchronies between visual cues and footfalls showed strong long-range persistence across all paced-walking conditions, with the PPS condition showing the strongest persistence among all conditions.

$SE = -0.2823 \pm 0.0398$, $t_{45} = -7.1017$, $P = 7.2009 \times 10^{-6}$, 95%CI = [$-0.3624$, $-0.2023$]). The effect of each pacing signal was consistent with previous reports where persistent visual cues accentuating the persistence in stride intervals and random visual cues attenuating the persistence in stride intervals observed during self-paced walking [119, 120].

Irrespective of whether the pacing signal was persistent or not, asynchronies between visual cues and footfalls showed persistence in each paced-walking condition (PPS: $M \pm SD\ H_{fGn} = 1.00 \pm 0.10$; $t_{9,2} = 13.8829$, $P = 2.2057 \times 10^{-7}$, 95%CI = [0.2077, 0.5664]; SPPS: $H_{fGn} = 0.86 \pm 0.22$; $t_{9,2} = 4.5783$, $P = 0.0013$, 95%CI = [0.1692, 0.4996]; GRPS: $H_{fGn} = 0.83 \pm 0.17$; $t_{9,2} = 5.5744$, $P = 3.4541 \times 10^{-4}$, 95%CI = [0.1920, 0.4543]; and URPS: $H_{fGn} = 0.77 \pm 0.09$; $t_{9,2} = 7.8018$, $P = 2.7032 \times 10^{-5}$, 95%CI = [0.1803, 0.3275]; Fig 6, **bottom**). The persistence in asynchrony across all paced-walking conditions is an early glimmer of challenges to the predictive modeling accounts of stride-to-stride control of stepping. Specifically, the predictive modeling accounts suggest entailing asynchronies between visual cues and footfalls that blend uncorrelated Gaussian noise and perturbation-contingent short-lag negative correlation. We find here that the asynchronies are negative and extend across long ranges instead of being short-lag and positive.

However, persistence is not necessarily more than a passing nuisance to modeling. Persistence fails to perturb the predictive modeling accounts of stride-to-stride control of stepping [6] because, on its own, even long-range persistence might be a strictly linear and so time-symmetric feature [48]. It is mathematically possible that such persistent signals are invariant across time [42]—that would be the way to preserve the predictive model. But this elaboration narrows the margin to fit a short-lag negative autocorrelation, and the predictive model might become more difficult to sustain if the long-range correlations exhibit nonlinearity in how they vary across time. Therefore, we next searched for the potential signature of nonlinearity in the observed variations in cue intervals, footfalls, and asynchronies between visual cues and footfalls.

## Asynchronies between visual cues and footfalls showed cascade-like intermittency

We know people adapt to continuous changes by making small, adaptive tunings and modifications. So, when participants adapt their actions to match their footfalls with visual cues, we should anticipate that the temporal structure of stride intervals will alter over time. Consequently, the persistence in stride intervals—and in asynchronies between visual cues and footfalls—might vary across time, resulting in an assortment of Hurst exponents. While $H_{fGn}$ represents the dominant persistent structure governing the entire time series, long-range correlations will inevitably wax and wane around this generalized $H_{fGn}$ value. Hence, we can estimate the multiple local fractal-scaling exponents $\alpha$ and construct the so-called "multifractal spectrum" whose width $\Delta\alpha$ indicates the diversity of long-range correlations in the same time series (Fig 7). Hence, the multifractal spectrum width gives us a first look at how flexibly an individual coordinates its footfalls across time while walking. We might even draw a conceptual analogy between $\Delta\alpha$ and $SD$—whereas $H_{fGn}$ gives an average-like summary description of long-range correlations, $\Delta\alpha$ is not unlike $SD$ or even a range insofar as it expresses the amount of variation around the average-like $H_{fGn}$.

Diversity in the temporal structure can reflect both linear and nonlinear correlations. In the former case, variation across time is homogeneous and time-symmetric, that is, the same sequence is governed by the same $H_{fGn}$ backward and forward. However, as the central limit theorem expects variation of the sample mean for samples of a given size around a population mean, we can expect the fractal-scaling exponents to vary with series length. Therefore, a portion of the multifractal spectrum represents uniform fluctuation across time and has a linear origin. In the latter case of nonlinear sources of heterogeneity, variation in temporal structure could reflect a progression from the start of the walking to its middle and finally to its conclusion. In this latter case, a varying temporal structure is not just sampling error but may reflect the structured accumulation of errors suggesting that the interaction of movements is spanning multiple timescales. Fortunately, we can use a $t$-statistic, $t_{MF}$—which we refer to as the "multifractal nonlinearity," to distill out two different parts of the multifractal spectrum width: the first predictable from the linear structure and the second revealing a nonlinear structure due to nonlinear interactions across timescales. Specifically, $t_{MF}$ takes the subtractive difference between $\Delta\alpha$ for the original time series and that for the 32 iterative amplitude-adjusted Fourier transform (IAAFT) surrogates, dividing by the standard error of $\Delta\alpha$ for the surrogates. IAAFT randomizes original values of the time series time-symmetrically around the autoregressive structure, generating surrogates having randomized phase ordering of the original time series' spectral amplitudes while preserving linear temporal correlations [133, 134]. $t_{MF} > 1.98$ is interpreted as evidence of nonlinear interactions across timescales as in an intermittent cascade model. Beyond strictly dichotomous treatment, greater $t_{MF}$ indicates stronger evidence of cascade-like intermittency.

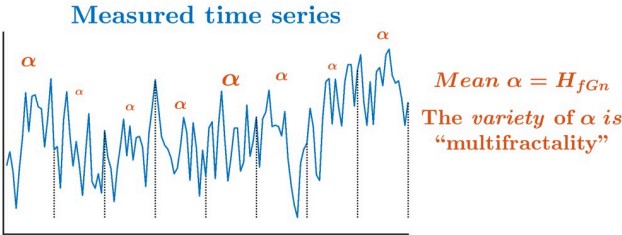

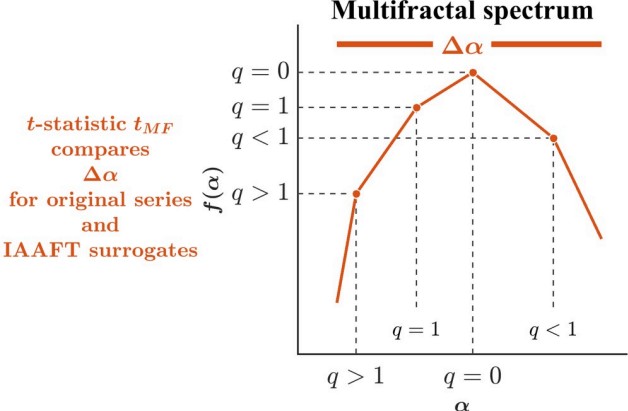

**Fig 7. Schematic portrayal of $\Delta_\alpha$ and $t_{MF}$.** While $H_{fGn}$ is the best single, average description of temporal structure for the whole time series, the fractal structure might change across time as indicated by local scaling exponents $\alpha$. $\Delta\alpha$ indicates the width of a spectrum of fractal exponents across time $f(\alpha)$ indicating how much of the time series exhibits each value of $\alpha$. The one-sample $t$-statistic tMF takes the subtractive difference between $\Delta\alpha$ for original time series and an average $\Delta\alpha$ for 32 surrogates, dividing by the standard error of $\Delta\alpha$ for the surrogates. $t_{MF} > 1.98$ is interpreted as evidence of nonlinear interactions across timescales resembling an intermittent cascade.

Stride intervals and asynchronies between visual cues and footfalls exhibited multifractality consistent with nonlinear correlations not otherwise reducible to the cue signals. Only the pink noise pacing signal contained nonlinear interactions across timescales indicative of cascade-like intermittency ($M \pm SD$ $t_{MF} = 26.43 \pm 13.54$; Fig 8, ***top***) as compared to no significant multifractal nonlinearity in the other cue signals (SPPS: $t_{MF} = -11.73 \pm 2.40$; GRPS: $t_{MF} = -10.07 \pm 9.72$; URPS: $t_{MF} = -3.90 \pm 4.99$). Stride intervals during the self-paced walking showed strong evidence of cascade-like intermittency ($t_{MF} = 37.33 \pm 34.52$), confirming the previous reports of multifractality in stride-to-stride variations [50–52]. Curiously, nonlinear interactions in the pink noise pacing signal did not affect nonlinear interactions in stride intervals ($t_{MF} = 37.26 \pm 18.15$, $B \pm SE = -0.0707 \pm 14.4550$, $t_{45} = -0.0049$, $P = 0.9961$, 95%CI = [−29.1840, 29.0420]; Fig 8, ***middle***). Instead, the task of synchronizing footfalls with uncorrelated pacing signals amplified the nonlinear interactions in stride intervals (SPSS: $t_{MF} = 67.14 \pm 50.69$, $B \pm SE = 46.8200 \pm 14.4550$, $t_{45} = 3.2391$, $P = 0.0023$, 95%CI = [17.7070, 75.9330]; GRPS: $t_{MF} = 84.15 \pm 32.94$, $B \pm SE = 34.4020 \pm 14.4550$, $t_{45} = 2.3800$, $P = 0.0216$, 95%CI = [5.2886, 63.5150]; and URPS ($t_{MF} = 71.73 \pm 35.91$, $B \pm SE = 29.8120 \pm 14.4550$, $t_{45} = 2.0625$, $P = 0.0050$, 95%CI = [0.6990, 58.9250]). In sum, synchronizing footfalls to cues with a temporal structure comparable to the SPW prompts no change in gait, but synchronizing footfalls to cues with less of the inherent structure of the SPW tax the movement system and prompt novel coordination across timescales, perhaps requiring new nonlinear interactions not needed for synchronizing to the SPW-like cues.

*Asynchronies between visual cues and footfalls showed cascade-like intermittency*

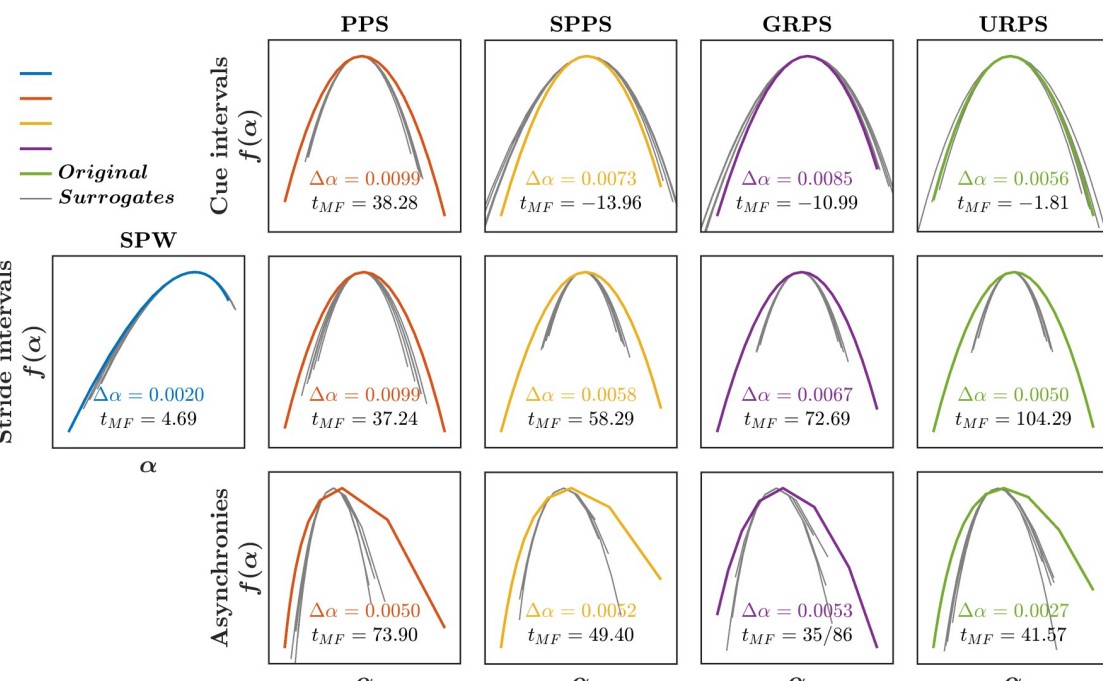

**Fig 8. Asynchrony between visual cues and footfalls showed cascade-like intermittency irrespective of intermittency in cue and stride intervals.** Multifractal spectra $f(\alpha)$ comprising a wide array of local fractal-scaling exponents, $\alpha$ across the series. The multifractal spectrum width, $\Delta\alpha$, encodes the heterogeneity of persistence in fluctuations in measurements across shorter and longer separations in time, in cue intervals (***top***), stride intervals (***middle***), and asynchrony between visual cues and footfalls (***bottom***) for a representative participant. Stride intervals during the self-paced walking (SPW) showed wider than surrogate spectrum—$t_{MF} >> 1.98$ (colored vs. grey lines), indicating strong nonlinear temporal correlations due to nonlinear interactions across timescales. Among the cue signals, only the pink noise pacing signal (PPS) exhibited nonlinear interactions across timescales, the corresponding stride intervals showed strong signatures of cascade-like intermittency indicated by wider than surrogate spectra. Hence, the SPPS, GRPS, and URPS could not fully break the nonlinearity in stride intervals. Contrary to the prediction of the linear relationship between following the predictive modeling accounts of stride-to-stride control of stepping, asynchronies between visual cues and footfalls showed cascade-like intermittency across all paced-walking conditions, most strongly in the PPS condition.

Nonlinear interactions across timescales were a generic feature of asynchronies between visual cues and footfalls across all cue signals irrespective of differences in nonlinear interactions (PPS: $M \pm SD\ t_{MF} = 73.58 \pm 33.78$; SPPS: $t_{MF} = 44.52 \pm 23.57$; GRPS: $t_{MF} = 42.57 \pm 31.98$; URPS: $t_{MF} = 31.58 \pm 20.96$; Fig 8, ***bottom***). The evidence of multifractality indicates variation in the long-range correlations across time beyond strictly linear temporal correlations. Practically, the time-symmetric possibility that linear models of power-law autocorrelation could leave each timescale unperturbed by variations at other timescales. To be clear: the strictly linear models of power-law autocorrelation would have allowed the predictive modeling to greet each new step with the same logic. Nonlinear interactions across scales indicate that each current step is the latest front in a cascade tumbling forward from the first steps. No step is comparable to the next, making short-lag responsivity an unreliable foundation for the stride-to-stride control of stepping. No single step occurs in a vacuum; instead, every step unfolds from the rich substrate of events that interact across many scales of time. Given this rich heredity wherein each step inherits the influence of steps at many time scales past, the control of gait may thus require less explicit intervention on each step. The coordination of each step unfolds implicitly through the ongoing accumulation of multiscaled events in gait. In this formulation,

*Linear and nonlinear measures of variability at a glance*

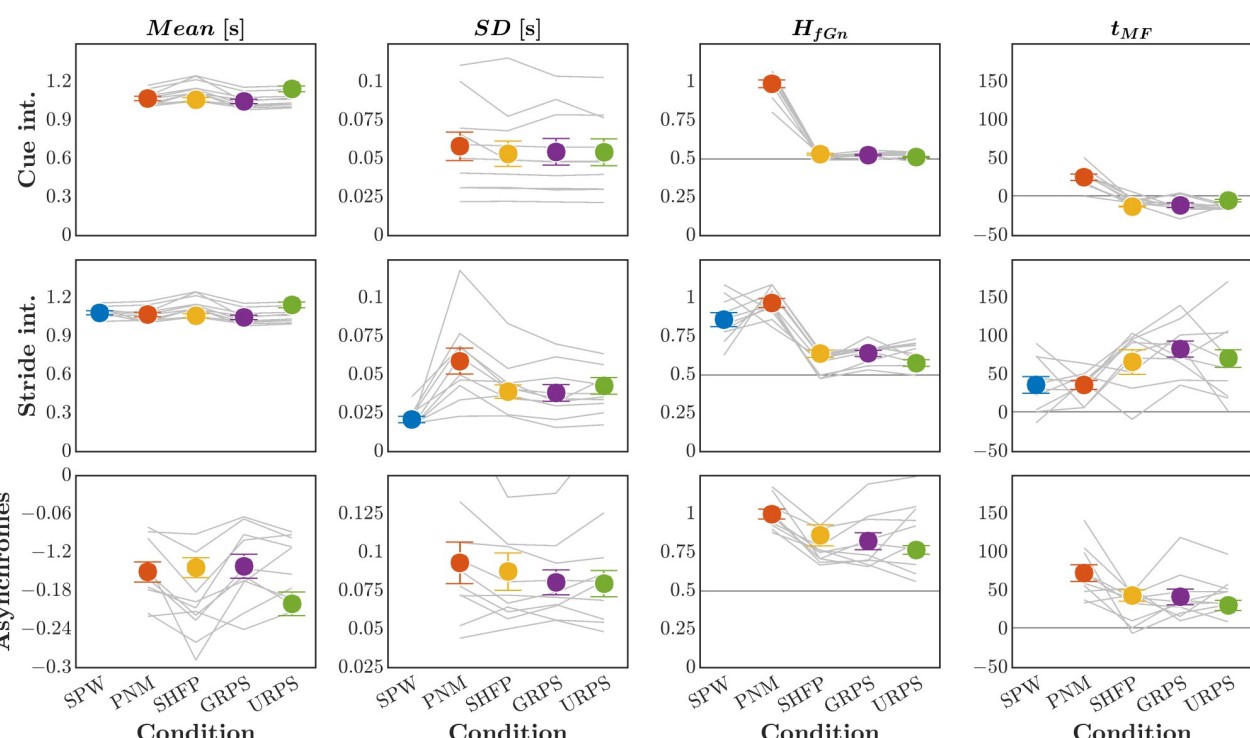

**Fig 9. Summary of the outcome variables.** $M \pm SEM$ values of linear and nonlinear measures of variability in cue intervals (***top***), stride intervals (***middle***), and asynchrony in the timings of visual cues and footfalls (***bottom***), across all participants ($N = 10$). Grey lines indicate values for individual participants.

the negative mean asynchrony is not a time-invariant outcome of time-invariant physiological mechanisms—instead, it is the time-varying coordination across timescales in ongoing gait variability.

Finally, we offer Fig 9 as a composite portrait of the linear and nonlinear features reviewed above.

## Stronger cascade-like intermittency was correlated with larger negative mean asynchrony

We had predicted that cascade processes in the motor sequence of asynchronies between visual cues and footfalls would facilitate anticipatory stepping (Hypothesis 3), that is, that individual trials' negative mean asynchrony might correlate specifically with corresponding trials' multi-fractal nonlinearity $t_{MF}$ of the asynchronies—and not with any simpler linear portrayal of the stride-to-stride variations. We examined the relationship between the mean negative asynchrony (between visual cues and footfalls) and (i) the magnitude of variation, quantified by the standard deviation $SD$; (ii) persistence in variation, quantified by the Hurst exponent $H_{fGn}$; and (iii) cascade-like intermittency of variation, quantified by the multifractal nonlinearity $t_{MF}$. We did not find any relationship between mean negative asynchrony with $SD$ or with $H_{fGn}$ of asynchronies in any of the four paced-walking conditions ($P$s all $>0.05$; Fig 10). Interestingly, mean negative asynchrony and $t_{MF}$ of asynchronies negatively correlated in response to the pink noise pacing signal (Pearson's $r = -0.8071$, $P = 0.0048$) and the Gaussian

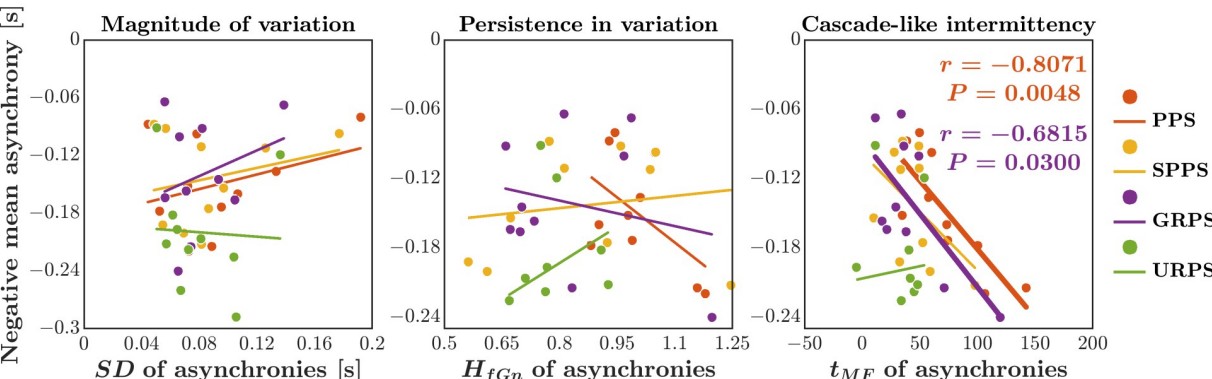

**Fig 10. Stronger cascade-like intermittency was correlated with larger negative mean asynchrony.** The relationships of mean negative asynchrony between visual cues and footfalls with the magnitude of variation, i.e., $SD$, of asynchronies (left), persistence, i.e., $H_{fGn}$, of asynchronies (center), and strength of cascade-like intermittency, i.e., $t_{MF}$, of asynchronies across all participants ($N = 10$). The negative mean asynchrony showed no relationship with the magnitude of variation and persistence in asynchronies but was significantly larger for a participant with a stronger cascade-like intermittency in asynchronies for walking in synchrony with the pink noise pacing signal (PPS) and Gaussian distributed random pacing signal (GRPS).

distributed random pacing signal ($r = −0.6815$, $P = 0.0300$) but not the other two conditions ($P$s > 0.05). These results bear our earlier points about the implications of cascade-like intermittency. The amount of negative mean asynchrony is not a fixed result from stable, time-invariant patterns of temporal correlations—instead, negative mean asynchrony is sensitive to the interactions unfolding over multiple timescales of the stride sequence. These correlations between multifractal nonlinearity $t_{MF}$ and negative mean asynchrony suggest anticipatory processes arising by cascade processes in the motor sequence.

## Discussion

The present results reveal evidence strongly at odds with the established explanations of gait control through predictive modeling. There were no predictable patterns in the empirical evidence related to walking and the cues that lead to negative mean asynchronies between cues and footfalls. We confirmed that self-paced walking broke ergodicity in the long run. The temporal organization of stride-to-stride variations during overground walking mimicked the ergodicity-breaking or ergodicity-preserving properties of the cue signal. However, the asynchronies that should be the primary focus of predictive modeling have none of the predictability. Across the board, no matter the ergodicity of the cue signal, asynchronies were ergodicity-breaking. This evidence undermines any notion that deviations in gait can be predictable. As stressed in the Introduction of this paper, the failure of representativity in ergodicity breaking amounts to a failure of any attempt to model and then predict. As if this ergodicity breaking were not problematic enough for the predictive-modeling account, all strides and asynchronies showed long-range correlations consistent with fractal scaling. This finding is at odds with the predictive-modeling expectations that stride-to-stride variations should be uncorrelated and that control should manifest as a short-lag negative autocorrelation. The fractal-like pattern of asynchronies varied across time, manifesting as a multifractal structure beyond what could be reduced to a linear structure. Hence, contrary to linear explanations of fractality in gait [6], the fractal structure of gait is fundamentally at odds with linearity and predominantly reflects the role of nonlinear interactions across time scales. However unpredictable as cues and asynchronies were, the nonlinear interactions across scales appeared to hold the key to how humans

could coordinate their gait with unpredictable cues: the estimable degree of multifractality attributable to nonlinearity correlated significantly with negative mean asynchrony. Hence, the negative mean asynchrony that prevailing accounts have repeatedly attributed to predictive modeling may reflect the role of cascading dynamics capable of coordinating events at multiple time scales. Gait control might extend beyond so-called "weak" anticipation prone to limitations of a predictive model and might rest on so-called "strong" anticipation founded in long-range, multi-scaled coupling amongst random fluctuation [104–107].

We tested three hypotheses about the predictability of deviations in gait and about its implications for anticipatory control of gait. First, we predicted that stride-to-stride variations would be nonergodic, especially in the unperturbed case of self-pacing; that is, individual stride intervals will not resemble the average of stride intervals over the long run (Hypothesis 1a). However, when we perturb gait by asking participants to entrain strides to visual cues having variable timing of presentation, we expected stride intervals to show ergodicity breaking (Hypothesis 1b) and the same for asynchronies between visual cues and footfalls (Hypothesis 1c), with less ergodicity breaking in response to cues with ergodic temporal structure (e.g., Gaussian or uniformly distributed uncorrelated noise, as well as shuffled pink noise; Hypothesis 1d). Second, we predicted that stride intervals and asynchronies between visual cues and footfalls would be fractal (Hypothesis 2a) and multifractal (Hypothesis 2b) rather than identically distributed, randomized, or independently sequenced noise with the same values and probability distribution. Third, we predicted that the multifractal evidence of nonlinearity in asynchronies between visual cues and footfalls would correlate with the negative mean asynchrony comparing footfalls to cues (Hypothesis 3). The results supported all hypotheses with two exceptions. First, the expected cue-sensitivity of ergodicity breaking made stride intervals ergodic in synchrony with the GRPS and URPS, and second, multifractal nonlinearity was not associated with negative mean asynchrony in the SPPS and URPS.

Our first hypothesis was primarily opening the question of whether the individual footfalls in gait are sufficiently representative, from one gait cycle to the next, to support a predictive model. We tested this hypothesis using the Thirumalai-Mountain (TM) metric—the gold standard in contemporary statistical physics for evaluating the degree of ergodicity breaking [10–13, 122]. A relatively coarse way to begin to understand ergodicity is as the equivalence of group averages with time averages [124, 135–138]. When these Thirumalai-Mountain metric estimates of ergodicity breaking for original and shuffled time series diverge from one another across time, the amount of divergence between them signifies how much the original time series breaks ergodicity.

Ergodicity breaking in gait was pervasive but sensitive to cue structure. For instance, self-paced walking (SPW) was only weakly nonergodic (Hypothesis 1a) and only across longer timescales. The cues we used exhibited a range of temporal correlations, one time series generated to exhibit fractional Gaussian noise (fGn), known in the engineering literature as "pink noise" [40]. This pink noise pacing signal (PPS) exhibited temporal correlations ($H_{fGn} > 0.5$), contrasting with the three types of uncorrelated cues signals: shuffled pink noise, Gaussian distributed randomized (i.e., exemplifying awGn), and uniformly distributed randomized pacing signals (SPPS, GRPS, and URPS, respectively). Each of these uncorrelated pacing signals removed specific parts of the fGn structure in PPS. The SPPS kept an originally fGn time series' probability distribution but decorrelated the sequence by shuffling. The GRPS had uncorrelated intervals with Gaussian probability distribution. Lastly, the URPS had uncorrelated intervals with uniform probability distribution. These three uncorrelated signals were intended to be indistinguishably different from PPS, and in fact, they all showed no difference of their $H_{fGn}$ from 0.5 or of their multifractal spectrum widths from that of their linear surrogates. Consistent with Hypothesis 1b, stride interval time series broke ergodicity even in

coordination with cues both for PPS and SPPS conditions. However, the cue-sensitivity expected in Hypothesis 1d was so strong as to imply that ergodic cues GRPS and URPS prompted stride intervals that showed no ergodicity breaking, showing near perfect overlap with the TM metric for shuffled versions. Temporal correlations in fGn may thus be so strong as to produce an excess of variance in the long run such that the resulting probability distribution breaks ergodicity even in the absence of temporal correlations. In any event, the ergodicity breaking in the cue signal is evidently sufficient to break ergodicity in stride intervals—even without temporal correlations. Hence, the weakly broken ergodicity in SPW might be more fragile in response to cueing than current theorizing has yet expected [10, 11, 13].

Asynchronies between visual cues and footfalls showed a more robust pattern of ergodicity breaking. That is, no matter the gait control system's capacity to make its strides ergodic (e.g., with GRPS and URPS), gait control leaves strong ergodicity breaking in the asynchronies between visual cues and footfalls for all kinds of temporal structures. The ergodicity breaking was weaker in the asynchronies relative to the GRPS than relative to the other cue signals—it was strongest in SPPS and URPS (e.g., see the flatter original $E_B$-vs.-$t$ curves). To the present study's major concern, and in support of Hypothesis 1c, it appears that errors in gait control during paced walking break ergodicity—more so than during SPW. Perhaps, the Gaussian probability distribution might preserve slightly more ergodicity than the uniform probability distribution and the probability distribution of shuffled pink noise. But the bigger point here is that the asynchronies between visual cues and footfalls exhibit a statistical structure beyond the known limits of computational prediction. SPW is already weakly ergodicity breaking, but enlisting the gait control in an explicit coordination task only brings the ergodicity breaking into stronger clarity in the pattern of asynchronies. Furthermore, increased cognitive could have contributed to reducing fractal profile in Yano et al. [139]. However, the strong multifractal evidence of nonlinearity and of ergodicity breaking limit the usefulness or generality of inventories of independent components or constraints whose effects may be genuine but fleeting or challenging to reproduce. Despite previous attempts to construe the $1/f$ form of strides as a linear process (e.g., [6]) submitting to the independent effects of independent constraints such as cognitive load, the present work confirms both that gait is more than just $1/f$ and more than just linear. The origins of the multifractal cascade are evident in the ergodicity-breaking dynamics of strides and asynchronies suggest espousing more stochastic notions of causation that do not always allow identification of independent effects [140].

No matter the unpredictability of errors in gait control, the present results indicated that asynchronies between visual cues and footfalls showed a rich patterning consistent with fractal and even multifractal structures. Stride intervals during SPW were fractally structured, replicating a long-known pattern of results [37, 38, 74, 81, 141–146]. In support of Hypothesis 2a and 2b, stride intervals and asynchronies between visual cues and footfalls showed strong evidence of fractal and multifractal structure in temporal correlations, confirmed by comparisons with shuffled and linear-surrogate time series, respectively. Much in the same way that stride-to-stride variations could match the ergodicity of the cue signal, asynchronies between visual cues and footfalls showed fractal evidence of persistence. These results also align with the known tendency of "complexity matching," that is, for movement complexity to match the complexity of the environmental stimulation [106, 147–150].

Whereas broken ergodicity thwarts predictive modeling, multifractal nonlinearity may offer gait support for anticipatory, prospective control robust to unpredictability. As we had noted above, multifractal nonlinearity has been associated with the ability of the movement system to resolve unpredictable perturbations [106, 114, 115, 117]. As it turned out, the PPS was the only cue signal with statistically significant evidence of multifractal nonlinearity. All other cue signals exhibited negligible temporal correlations, and any variation in temporal

correlations they showed was indistinguishable from linearity. Our final question had been whether the gait control might build its anticipatory, prospective control on the foundation of multifractal nonlinearity in its deviations from planned behavior. So, Hypothesis 3 predicted that the anticipation of the cue would manifest as a correlation of multifractal nonlinearity with more negative mean asynchrony. Specifically, a negative association in which greater multifractal nonlinearity would correlate with better ability to step ahead of the cue onsets. Specifically, we tested the correlation of negative mean asynchrony with standard deviation ($SD$), fractally correlated persistence ($H_{fGn}$), and multifractal nonlinearity ($t_{MF}$) for each cue signal. We found a negative correlation specifically for multifractality nonlinearity and negative mean asynchrony with cueing signals PPS and GRPS but not the SPPS and URPS. The correlations in PPS and GRPS conditions thus support Hypothesis 3. We suspect that the failure of a correlation between multifractal nonlinearity and negative mean asynchrony in PPS and GRPS conditions reflects more of a failure of the negative mean asynchrony than of the relevance of multifractal nonlinearity. Specifically, we mean that the ergodicity breaking pattern of the asynchronies in the PPS and GRPS cases indicates stronger ergodicity (e.g., steeper slopes of the $E_B$-vs.-$t$ curves) than in the SPPS and URPS cases. Hence, the mean we might take of the asynchronies is demonstrably less stable in the SPPS and URPS cases and more stable in the PPS and GRPS cases.

## Conclusions and future directions

We have used "control" to describe persistence and cascade-like interactivity in stride-to-stride variations and absolute errors in the timings of visual cues and footfalls. We invoke this concept of "control" less as the straightforward task of timing individual strides following individual cues and more so as the implicit capacity of cascading properties—sometimes of experimental stimulation but also prospective organisms—to constrain ongoing action. We provide strong evidence supporting that the ergodicity-breaking, multifractal-like cascade structure of walking thwarts the expectations of the predictive modeling account of walking.

Although recognizing negative mean asynchrony despite sensory delays has motivated the forward model, the same empirical background more fully points toward the cascade-dynamical explanation, considering that, in his seminal studies on asynchronies, Dunlap [71] noticed that the errors tend to grow larger and larger in each direction (too early or too late) until a correction causes a change in direction, and then the same pattern repeats. Dunlap attributed this "drifting" to a frequency mismatch between stimulus and response. Since then, several studies using synchronization tasks have found that the 1/$f$ framework is more appropriate than the simple self-correcting models for describing the temporal correlations of the error series [151]. Ergodicity breaking implies not only that gait cycles are not comparable in the long-range but also that the asynchronies do not converge to a stable average. So, the CNS could usefully interpret this error simply through magnitude. The finding of persistence and cascade-like intermittency in stride-to-stride errors during paced walking indicates that a higher priority for the CNS might be the capacity of errors to evolve through interactions among multiple timescales.

We have provided a theoretical roadmap to study the statistical aspects of movement coordination and then use cascade-dynamical parameters that match these movement structures. This research framework could help identifying the relative contributions of feedforward and feedback control in post-perturbation motor outputs during and after locomotor performance following degrading sensory feedback [152]. The methods we used to assess ergodicity breaking (the Thirumalai-Mountain metric), persistence ($H_{fGn}$, computed using the detrended fluctuation analysis), and cascade-like intermittency ($t_{MF}$, computed using the multifractal

analysis) can also help investigate altered sensorimotor control of locomotion in older adults and clinical populations. For instance, these methods might be suitable for analyzing differences in young adults, older adults, and adults with neurodegenerative conditions that might affect walking. For instance, a lack of persistence in stride intervals during walking is characteristic of aging [39, 126] and Parkinson's disease [127, 153]. Paced walking with irregular metronomes embedded with statistical properties found in healthy populations has been shown to increase the persistence in stride intervals in older adults [154] and Parkinson's patients [155–157]. Is the loss of persistence in stride intervals in these groups also accompanied by a loss of cascade-like intermittency in the absolute errors in the timing of cues and footfalls? Does the entrainment to irregular metronomes merely restore patterns of persistence in stride intervals or also restore intermittency in the sensorimotor control of heel-strike timing? This knowledge can be applied to creating more effective rehabilitation devices that directly influence the sensorimotor control of walking and restore healthy walking in older adults and clinical populations.

## Materials and methods

### Ethics statement

Each participant gave informed written consent with full knowledge of the study objectives and details of the experimental procedure. The Institutional Review Board of the University of Nebraska Medical Center approved the present study (IRB # 511–16-EP) in accordance with the Declaration of Helsinki. All data were collected between January and May 2019 and fully anonymized before we accessed them.

### Participants

Thirteen healthy young adults with no self-reported neurological or musculoskeletal disorders voluntarily participated in the exchange of monetary reward. Participants who had to be reminded more than once per trial to match the timings of the right heel strike and that of the moving bar touching the stationary bottom bar were deemed not to follow the instructions and were excluded. Following this exclusionary criterion, 10 participants (3 women; $M \pm SD$ age: 24.8 ± 3.9 years) were included in the analysis.

### Experimental setup and procedure

Upon arrival to the laboratory, the participants were fitted with Footswitch FSR SmartLead$^{TM}$ sensors (Noraxon, Scottsdale, USA) under both heels to identify footfalls with a sampling frequency of 1500 Hz. This sampling frequency allowed the detection of footfalls with <1 ms precision. The participants completed five overground walking trials on a 200 m indoor track (Fig 2) with 5 min rest in-between, with each trial including a minimum of 700 strides (approximately 13-min duration). The participants were not forced to change their direction along the track. The curved nature of the track naturally led the participants to change direction during their walk.

   In the first trial, the participants were instructed to walk at a self-selected preferred pace, which we called self-paced walking (SPW). The next four trials required participants to time their right footfalls with oscillations of a horizontal bar on a mini HDMI video screen mounted on eyeglass frames. Footswitch sensors placed under both heels identified heel strike events. The timing of the oscillations in these four trials had pink noise pacing signal (PPS), shuffled pink noise pacing signal (SPPS), Gaussian distributed random pacing signal (GRPS), and uniformly distributed random pacing signal (URPS). SPPS had the same probability distribution

of values as PPS but lacked any temporal correlations. Both kinds of random pacing signals lack any temporal correlations but have different probability distributions of values. The four pacing signals were generated using custom scripts in MATLAB 2020b (MathWorks Inc., Natick, MA). The PPS signal was created iteratively by first simulating pink Gaussian noise using the function `pinknoise()`. Next, the noise was checked using the detrended fluctuation analysis (DFA; see below) to ensure it had an $H_{fGn}$ close to 1. If not, the process was repeated until convergence was met ($0.996 < H_{fGn} < 1.004$). PPS was shuffled to obtain SPPS. GRPS was created iteratively by randomly permuting the PPS signal until DFA yielded $0.496 < H_{fGn} < 0.504$. URPS was generated using the function `rand()` but also in an iterative fashion, checking for the same convergence criteria as in GRPS. The individual mean stride-to-stride interval and standard deviation observed during the self-paced walking trial were used to scale the four pacing signals for each participant. The order of the paced trials was pseudorandomized for each participant.

The participants wore a pair of non-prescription glasses in which the visual pacing cues were displayed on a mini HDMI screen (Vufine Inc., Sunnyvale, CA, USA; Fig 3). The visual cues involved a horizontal bar moving vertically between two stationary bars. The bar maintained a constant velocity throughout each desired gait cycle. The bar's velocity was derived from the average velocity of each participant, as determined during the self-paced walking trial. Subsequently, fluctuations were introduced, superimposing upon this mean velocity, and their characteristics were based on the selected distribution (e.g., pink, white, etc.). That being said, the bar's velocity could be informative for the participants, allowing them to predict the next instant of the footfall. For instance, let us consider two stride intervals, T1 and T2. If T1 is greater than T2, and the bar consistently moves at this constant velocity, it will cover a greater distance in T2 compared to T1. The participants were instructed to match their right footfalls to the instant the moving bar reached the stationary top bar and to match their left footfalls to the instant the moving bar reached the stationary bottom bar.

## Data processing

We processed all data in Matlab using custom scripts. Stride-to-stride intervals were calculated as the time between two consecutive footfalls of the same foot. Standard deviation (*SD*) quantified the variation in stride-to-stride intervals. Whenever feasible, the analysis focused on time series comprising 624 intervals, each representing 624 strides, though a few trials may have had fewer strides.

## Estimating ergodicity breaking

We investigated the ergodic properties of the visual cue intervals, footfalls, and footfall and cue asynchronies. Ergodicity refers to the convergence of the finite-ensemble average and the finite-time average. The finite ensemble is

$$\langle x_i(t) \rangle_N = \frac{1}{N} \sum_{i=1}^{N} x_i(t), \tag{1}$$

where $x_i(t)$ is the $i$th of $N$ realizations of cue intervals, footfalls, or footfall and cue asynchronies included in the average. In contrast, the finite-time average is

$$\overline{x_{\Delta t}} = \frac{1}{\Delta t} \int_{t}^{t+\Delta t} x(t)dt, \tag{2}$$

When the measured behavior $x$ changes at $T = \Delta t/\delta t$ discrete times $t + \delta t$, $t + 2\delta t$, . . ., the

finite-time average is

$$\overline{x_{\Delta t}} = \frac{1}{T \delta t} \sum_{\tau=1}^{T} x(t + \tau \delta t).$$

So the traditional definition of ergodicity is an equivalence between these two averages,

$$\lim_{\Delta t \to \infty} \frac{1}{\Delta t} \int_{t}^{t+\Delta t} x(t) dt = \lim_{N \to \infty} \frac{1}{N} \sum_{i=1}^{N} x_i(t).$$

We quantified the ergodicity of each time series of cue intervals, stride intervals, and asynchronies between visual cues and footfalls using a dimensionless statistic of ergodicity breaking $E_B$, known as the Thirumalai-Mountain metric [121, 122]. $E_B$ is an inverse metric of ergodicity, reflecting how the long-range persistence failure of variance to converge, in effect, undermines (or "breaks") the representativity of a sample, thereby undermining ergodicity. This $E_B$ statistic subtracts the squared total-sample variance, $\langle \overline{\delta^2(x(t))} \rangle^2$, from the average squared subsample variance, $\langle [\overline{\delta^2(x(t))}]^2 \rangle$, and divides the resultant by the squared total-sample variance, $\langle \overline{\delta^2(x(t))} \rangle^2$:

$$\text{EB}(x(t)) = \frac{\langle [\overline{\delta^2(x(t))}]^2 \rangle - \langle \overline{\delta^2(x(t))} \rangle^2}{\langle \overline{\delta^2(x(t))} \rangle^2}, \tag{3}$$

where $\delta^2(x(t))$ is sample-to-sample variance and this relationship is effectively the variance of sample variance divided by the squared total-sample variance. Rapid decay of $E_B$ to 0 for progressively larger samples, that is, $E_B \to 0$ as $t \to \infty$ implies ergodicity. Slower decay indicates less ergodic systems in which trajectories are less reproducible, and no decay or convergence to a finite asymptotic value indicates strong ergodicity breaking [9]. $E_B$-vs.-$t$ curves thus allow testing whether a given time series fulfills ergodic assumptions or breaks ergodicity and the extent to which it breaks ergodicity. We used $\Delta = 4$ for computing $E_B(x(t))$.

### Assessing the strength of persistence using detrended fluctuation analysis

Detrended fluctuation analysis (DFA) computes the Hurst exponent, $H_{fGn}$, quantifying the strength of long-range correlations [129, 130] for each time series of cue intervals, footfalls, and footfall and cue asynchronies, using the first-order integration of $T$-length time series $x(t)$:

$$y(i) = \sum_{k=1}^{i} (x(k) - \overline{x(t)}), \quad i = 1, 2, 3, \cdots, T, \tag{4}$$

DFA computes root mean square ($RMS$; i.e., averaging the residuals) for each linear trend $y_n(t)$ fit to $N_n$ non-overlapping $n$-length bins to build a fluctuation function:

$$f(v, n) =$$
$$\sqrt{\frac{1}{N_n} \sum_{v=1}^{N_n} \left( \frac{1}{n} \sum_{i=1}^{n} (y((v-1)\, n + i) - y_v(i))^2 \right)}, \quad n = \{4, 8, 12, \cdots\} < T/4. \tag{5}$$

$f(n)$ is a power law,

$$f(n) \sim n^{H_{fGn}}, \tag{6}$$

where $H_{fGn}$ is the scaling exponent estimable using logarithmic transformation:

$$\log f(n) = H_{fGn} \log n. \tag{7}$$

Higher $H_{fGn}$ corresponds to stronger long-range correlations. A time series with $H_{fGn}$ = 0.5 can be considered as having a spectrum representing white noise. In contrast, a time series with $H_{fGn}$ = 1 can be considered as having a spectrum representing pink noise.

## Accessing cascade-like intermittency using multifractal analysis

The current cascade-dynamical interest in nonlinear relationships among hierarchically nested timescales is beyond the scope of what $H_{fGn}$ can encode [47, 58, 158]. Beyond strictly linear temporal correlations, the nonlinearity of interactions across timescales implies one fractional scaling exponent and multiple scaling exponents. Hence, a thorough investigation of cascade-like intermittency requires generalizing the test for fractal scaling into multifractal modeling [59, 159].

Chhabra and Jensen's [160] direct method estimates multifractal spectrum width $\Delta \alpha$ for each time series of cue intervals, footfalls, and footfall and cue asynchronies by sampling a series $x(t)$ at progressively larger scales using the proportion of signal $P_i(n)$ falling within the $v$th bin of scale $n$ as

$$P_v(n) = \frac{\sum\limits_{k=(v-1)\,n+1}^{v \cdot N_n} x(k)}{\sum x(t)}, \quad n = \{4, 8, 16, \cdots\} < T/8, \tag{8}$$

As $n$ increases, $P_v(n)$ represents a progressively larger proportion of $x(t)$,

$$P(n_v) \propto n^{\alpha_v}, \tag{9}$$

whereby each $v$th bin may show a distinct relationship of $P(n)$ with $n$. The multifractal spectrum width indicates the heterogeneity of these relationships [161, 162].

Chhabra and Jensen's [160] method estimates $P(n)$ for $N_n$ non-overlapping bins of $n$-sizes and transforms them into a "mass" $\mu(q)$ using a $q$ parameter emphasizing higher or lower $P(n)$ for $q > 1$ and $q < 1$, respectively, in the form

$$\mu_v(q, n) = \frac{[P_v(n)]^q}{\sum\limits_{j=1}^{N_n}[P_j(n)]^q}. \tag{10}$$

Then, $\alpha(q)$ belongs to the multifractal spectrum only when the Shannon entropy of $\mu(q, n)$ scales with $\mu(q)$ estimated as

$$\begin{aligned} \alpha(q) \quad &= -\lim_{N_n \to \infty} \frac{1}{\ln N_n} \sum_{v=1}^{N_n} \mu_v(q, n) \ln P_v(n) \\ &= \lim_{n \to 0} \frac{1}{\ln n} \sum_{v=1}^{N_n} \mu_v(q, n) \ln P_v(n). \end{aligned} \tag{11}$$

Each estimated value of $\alpha(q)$ belongs to the multifractal spectrum only when the Shannon

entropy of $\mu(q, n)$ scales with $n$ according to the Hausdorff dimension $f(q)$ [160], where

$$
\begin{aligned}
f(q) &= -\lim_{N_n \to \infty} \frac{1}{\ln N_n} \sum_{\nu=1}^{N_n} \mu_\nu(q, n) \ln \mu_\nu(q, n) \\
&= \lim_{\nu \to 0} \frac{1}{\ln n} \sum_{\nu=1}^{N_n} \mu_\nu(q, n) \ln \mu_\nu(q, n).
\end{aligned}
\tag{12}
$$

For values of $q$ yielding a strong relationship between Eqs (11) & (12)—in this study, correlation coefficient $r > 0.975$, the parametric curve $(\alpha(q), f(q))$ or $(\alpha, f(\alpha))$ constitutes the multifractal spectrum and $\Delta\alpha$ (i.e., $\alpha_{max} - \alpha_{min}$) constitutes the multifractal spectrum width. $r$ determines determines that only scaling relationships of comparable strength can support the estimation of the multifractal spectrum.

## Surrogate testing using Iterated Amplitude-Adjusted Fourier Transformation (IAAFT) generated $t_{MF}$

Multifractality is a necessary entailment of cascade-like intermittency, but evidence of multifractality alone is insufficient to diagnose cascade-like intermittency. Multifractality can follow from sources other than nonlinear interactions across timescales [163]. To identify whether non-zero multifractal spectrum width $\Delta\alpha$ reflected multifractality due to nonlinear interactions across timescales, $\Delta\alpha$ for the original cue intervals, footfalls, and asynchrony between visual cues and footfalls was compared to $\Delta\alpha$ for 32 IAAFT surrogates [133, 134]. IAAFT randomizes original values of the time series time-symmetrically around the autoregressive structure, generating surrogates with randomized phase ordering but preserving the original time series' amplitude spectrum (a detailed step-by-step guide to surrogate testing provided in Kelty-Stephen et al. [113]. The one-sample $t$-statistic, $t_{MF}$—which we have referred to as the "multifractal nonlinearity," took the subtractive difference between $\Delta\alpha$ for the original time series and that for the 32 surrogates, dividing by the standard error of $\Delta\alpha$ for the surrogates.

## Statistical analysis

We used separate paired-sample $t$-tests to compare the strength of persistence between the original time series of cue intervals, stride intervals, and asynchronies between visual cues and footfalls and their respective shuffled counterparts for self-paced walking and the four paced-walking conditions. We used two separate linear mixed-effects models to examine the influence of paced-walking conditions on the strength of persistence ($H_{fGn}$) and cascade-like intermittency ($t_{MF}$) in stride intervals. We accounted for individual differences by introducing a random effect of participant identity in the linear mixed-effects analysis. Finally, we used Pearson's correlation tests to examine the relationship between the negative mean asynchrony in the timing of visual cues and footfalls and (i) the magnitude of variation, quantified by the standard deviation SD; (ii) persistence in variation, quantified by the Hurst exponent $H_{fGn}$; and (iii) cascade-like intermittency, quantified by the multifractal nonlinearity $t_{MF}$. We performed all statistical analyses in MATLAB 2020b and considered the outcomes significant at the two-tailed alpha level of 0.05.

## Supporting information

**S1 Dataset. Raw data analyzed in the present study.** The original time series of cue intervals, stride intervals, and asynchronies between visual cues and footfalls for self-paced walking and

the paced walking conditions (units in seconds).
(XLSX)

## Author Contributions

**Conceptualization:** Madhur Mangalam, Damian G. Kelty-Stephen, Nick Stergiou, Aaron D. Likens.

**Data curation:** Madhur Mangalam, Joel H. Sommerfeld.

**Formal analysis:** Madhur Mangalam.

**Funding acquisition:** Nick Stergiou.

**Methodology:** Madhur Mangalam, Joel H. Sommerfeld.

**Supervision:** Madhur Mangalam, Nick Stergiou, Aaron D. Likens.

**Visualization:** Madhur Mangalam.

**Writing – original draft:** Madhur Mangalam, Damian G. Kelty-Stephen.

**Writing – review & editing:** Madhur Mangalam, Damian G. Kelty-Stephen, Joel H. Sommerfeld, Nick Stergiou, Aaron D. Likens.

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
