## [Decision Letter · Decision Letter 0]

20 Jul 2023

PONE-D-23-13654Temporal organization of stride-to-stride variations contradicts predictive models for sensorimotor control of walkingPLOS ONE

Dear Dr. Mangalam,

Thank you for submitting your manuscript to PLOS ONE. After careful consideration, we feel that it has merit but does not fully meet PLOS ONE’s publication criteria as it currently stands. Therefore, we invite you to submit a revised version of the manuscript that addresses the points raised during the review process.

ACADEMIC EDITOR:Overall, the paper is a well organized study with clear research question and it touches interesting points. In any case, an eminent reviewer highlighted important points that have to be carefully responded, for this reason Authors can proceed by carefully addressing the comments reported in the decision letter.

We look forward to receiving your revised manuscript.

Kind regards,

Andrea Tigrini, Ph.D.

Academic Editor

PLOS ONE

3. We note that Figures 1 and 2 in your submission contain copyrighted images. All PLOS content is published under the Creative Commons Attribution License (CC BY 4.0), which means that the manuscript, images, and Supporting Information files will be freely available online, and any third party is permitted to access, download, copy, distribute, and use these materials in any way, even commercially, with proper attribution. For more information, see our copyright guidelines: http://journals.plos.org/plosone/s/licenses-and-copyright.

a. You may seek permission from the original copyright holder of Figures 1 and 2  to publish the content specifically under the CC BY 4.0 license.

Additional Editor Comments:

Overall, the paper is a well organized study with clear research question and it touches interesting points. In any case, an eminent reviewer highlighted important points that have to be carefully responded, for this reason Authors can proceed by carefully addressing the comments reported in the decision letter.

Reviewers' comments:

Reviewer's Responses to Questions

**Comments to the Author**

1. Is the manuscript technically sound, and do the data support the conclusions?

Reviewer #1: Yes

2. Has the statistical analysis been performed appropriately and rigorously? 

Reviewer #1: Yes

3. Have the authors made all data underlying the findings in their manuscript fully available?

Reviewer #1: No

4. Is the manuscript presented in an intelligible fashion and written in standard English?

Reviewer #1: Yes

5. Review Comments to the Author

Reviewer #1: General comments:

This study analyzed gait cycle variability during self-paced and visual-cue-paced walking. Variability of gait cycle and associated time series were characterized for (i) visual cues (visual metronome), (ii) footfalls (gat cycle), and (iii) asynchronies between visual cues and footfalls by using several metrics, including: (1) ergodicity breaking based on the Thirumalai-Mountain metric, (2) Hurst exponent based on the standard DFA, (3) singularity spectrum (multi-fractal spectrum) based on Chhabra and Jensen’s method. Results of the analysis were novel and interesting. Thus, the manuscript deserves publication. However, I have major concerns on the interpretation of the results. Indeed, the interpretation of the authors is well summarized in the title of the manuscript: "Temporal organization of stride-to-stride variations contradicts predictive models for sensorimotor control of walking." To be brief, I don't think the predictive models of human gait has not been gained popularity (although there might be many such models in the field of robotics for bipedal robot locomotion), unlike those for motor control of upper limb voluntary movements. Thus, the term "contradiction" to non-validated predictive models sounds slightly odd, at least for me. Although I am hopeful that I would be convinced by the rebuttal arguments, I recommend to lower the tone of statement in the revised manuscript.

Major comments:

- As mentioned above, the gait and posture are classified as the automatic movement, and they are substantially different from voluntary movements. On the other hand, the visual metronome conditions used in this study provide a continuous guide for the timing of forthcoming footfall, which might make such a gait more or less voluntary, compared to the natural SPW. The current study (particularly the interpretation of the results) is mixed up those two different kinds of motor control, and derived a conclusion in a unified manner about the sensorimotor control of gait in general. Since the major outcomes on the ergodicity breaking (Fig. 4 bottom), the scale-free like behaviors in DFA (Fig. 6 bottom), and cascade-like intermittency in singularity spectrum (Fig. 8 bottom) are all for the asynchrony between visual cues and footfalls, and the asynchrony is heavily dependent on the voluntary decision making on the foot placement, the assertion of the contradiction to the predictive control should be limited to such a gait pattern with the voluntary attempt to regulate the footfall timings, not necessarily to the gait control, including the natural SPW in general. I think the title of this paper should be weaken in this way. For example, "Temporal organization of stride-to-stride variations contradicts predictive models for sensorimotor control during metronomed (or volitional) walking."

- The other concern is also related to the interpretation of the authors about the predictive model. Particularly, rationale to derive their conclusion from the ergodicity breaking. It seems that the authors consider the predictive control and the ergodicity of the gait cycle variability equivalent. Can it be justified theoretically? In my opinion, simple interventions to the gait rhythm, which are determined (generated) based on the state of the gait control system only for the present (i.e., only for the past one cycle) might be able to generate long-range correlation in the gait cycle variability. Such simple but nonlinear (and/or impulsive) interventions, for example, might include the phase resetting and the intermittent control that utilizes a stable manifold of unstable limit cycle of the gait system with no active feedback control. See Fu et al (Biological Cybernetics 114 (1), 95-111, 2020), for example. Note that the intermittent control in this case is not activated by a threshold-crossing that leads to the risk of fall, but the active feedback controller is switched off when the state of the system visits a neighbor of the stable manifold, by which the state point of the system might exhibit a slow sliding motion along the stable manifold in a manner of stride-to-stride basis (a transiently converging sequence of the state points in terms of Poincare mapping that observes the state point at every switching-off event). Note that my point is not the details of the nonlinear feedback control, but the gait fractality (and possibly ergodicity breaking phenomena) could be emerged through some control mechanisms (such as the phase resetting and the intermittent controller, either predictive or non-predictive) that modulate the gait rhythm based on the state of the gait control system only for the present cycle. If so, ergodicity breaking does not necessarily conflict with the models about the sensorimotor control that correct the timings of the footfall from one stride to the next. This is why I would not be able to agree with the interpretation of the authors.

- There are some studies (e.g., Fluctuation of stride time intervals during walking with smartphone, S Yano, L Dimalanta, Y Suzuki, T Nomura, 2019 IEEE 1st Global Conference on Life Sciences and Technologies, DOI:10.1109/LifeTech.2019.8884072) showing that the cognitive load decreases the persistency in gait cycle variability. From Fig. 6 (middle), I can see a similar tendency for SPPS, GRPS and URPS. However, since the positive persistency is lost also in the visual metronome, the decrease in the persistency for the gait cycle variability could be caused by the metronome. Please discuss possible causes of the loss of the fractality in the gait cycle variability.

Minor comments:

Method.

- Please elaborate more about how the horizontal bar exhibited as the visual cues moved. Does the bar move with a constant velocity for each desired gait cycle? That is, for example, let me consider two stride intervals T1 and T2. If T1>T2, and if the bar moves at a constant velocity, it should move faster for T2 than for T1. In this case, the velocity of the bar is very informative for the subject, by which one can predict next instant of the footfall. Is this ok? Did you provide such information to the subject on purpose?

- Was the subject forced to change the direction (maybe slightly) along the track field? Did the subject walk straight? I am asking this because the walk way shown in Fig. 2 is curved.

- Fig. 2: Please do not call the stride between two footprints as "stride interval" since the stride interval is reserved for the stride time interval.

- l292: (ii) should be (iii)

-Fig. 3 top traces: Why did the graphs of cues appear with the artifact of digitization? The resolution of the graphs should be much higher. Was such low resolution visual metronome data used in the experiment?

- Fig. 7: I don't think the illustration of multi-fractal (distributed singularity along time axis) is correct rigorously speaking, although it might be intuitively helpful for some people who are not familiar with the concept. In reality, different strength of singularity for multiple scales are distributed in a nested manner. I recommend to remove this part of Fig. 7, or make it more mathematically sound.

- E_B: Please provide a clear definition of ¥delta(x(t)) both in the main text and in the Method section.

6. PLOS authors have the option to publish the peer review history of their article (what does this mean?). If published, this will include your full peer review and any attached files.

Reviewer #1: No

---

## [Author Response · Author response to Decision Letter 0]

25 Jul 2023

We have responded to reviewers in a PDF file attached with the submission.

---

## [Decision Letter · Decision Letter 1]

4 Aug 2023

Temporal organization of stride-to-stride variations contradicts predictive models for sensorimotor control of footfalls during walking

PONE-D-23-13654R1

Dear Dr. Mangalam,

We’re pleased to inform you that your manuscript has been judged scientifically suitable for publication and will be formally accepted for publication once it meets all outstanding technical requirements.

Kind regards,

Andrea Tigrini, Ph.D.

Academic Editor

PLOS ONE

Additional Editor Comments (optional):

Authros have solved all the comments previuosly provided by the Expert. The paper can be accepted

Reviewers' comments:

Reviewer's Responses to Questions

**Comments to the Author**

1. If the authors have adequately addressed your comments raised in a previous round of review and you feel that this manuscript is now acceptable for publication, you may indicate that here to bypass the “Comments to the Author” section, enter your conflict of interest statement in the “Confidential to Editor” section, and submit your "Accept" recommendation.

Reviewer #1: All comments have been addressed

2. Is the manuscript technically sound, and do the data support the conclusions?

Reviewer #1: Yes

3. Has the statistical analysis been performed appropriately and rigorously? 

Reviewer #1: Yes

4. Have the authors made all data underlying the findings in their manuscript fully available?

Reviewer #1: Yes

5. Is the manuscript presented in an intelligible fashion and written in standard English?

Reviewer #1: Yes

6. Review Comments to the Author

Reviewer #1: (No Response)

7. PLOS authors have the option to publish the peer review history of their article (what does this mean?). If published, this will include your full peer review and any attached files.

Reviewer #1: No

---

## [Editor Report · Acceptance letter]

14 Aug 2023

PONE-D-23-13654R1 

Temporal organization of stride-to-stride variations contradicts
predictive models for sensorimotor control of footfalls during
walking 

Dear Dr. Mangalam:

I'm pleased to inform you that your manuscript has been deemed suitable for publication in PLOS ONE. Congratulations! Your manuscript is now with our production department. 

Kind regards, 

on behalf of

Dr. Andrea Tigrini 

Academic Editor

PLOS ONE